# DIVERSE GENOMIC EMBEDDING BENCHMARK FOR FUNCTIONAL EVALUATION ACROSS THE TREE OF LIFE

## ABSTRACT

Biological foundation models hold significant promise for deciphering complex biological functions. However, evaluating their performance on functional tasks remains challenging due to the lack of standardized benchmarks encompassing diverse sequences and functions. Existing functional annotations are often scarce, biased, and susceptible to train-test leakage, hindering robust evaluation. Furthermore, biological functions manifest at multiple scales, from individual residues to large genomic segments. To address these limitations, we introduce the Diverse Genomic Embedding Benchmark (DGEB), inspired by natural language embedding benchmarks. DGEB comprises six embedding tasks across 18 expert curated datasets, spanning sequences from all domains of life and encompassing both nucleic acid and amino acid modalities. Notably, four datasets enable direct comparison between models trained on different modalities. Benchmarking protein and genomic language models (pLMs and gLMs) on DGEB reveals performance saturation with model scaling on numerous tasks, especially on those with underrepresented sequences (e.g. Archaea). This highlights the limitations of existing modeling objectives and training data distributions for capturing diverse biological functions. DGEB is available as an open-source package with a public leaderboard at `URLhiddenforanonymity`.

## 1 INTRODUCTION

Biological sequences encode complex molecular, evolutionary and biophysical information that govern biological function. Deep learning models have been proposed as promising methods for extracting biologically relevant functional information from sequence data. The promise of "biological foundation models" enabling functional interpretation of sequences has resulted in many modeling efforts in protein (Rives et al., 2021; Madani et al., 2023; Elnaggar et al., 2022) and genomic (Dalla-Torre et al., 2023; Hwang et al., 2024; Nguyen et al., 2024) sequence modalities. While the field has seen major advances in AI-enabled structure prediction of protein sequences (Jumper et al., 2021; Baek et al., 2021), validated successes for AI-enabled function prediction remain limited (Li et al., 2024). Slow progress in function prediction of sequences can be attributed to the following main challenges:

1. **Unlike for structural prediction tasks, objective measurements of function do not exist.** Structure prediction tasks benefit from objective evaluation metrics based on quantifiable atomic distances (Mariani et al., 2013). However, biological function is inherently multifaceted and context-dependent, making direct quantitative assessment difficult.

2. **Functional labels are sparse, biased, and prone to leakage.** Labels are heavily biased towards model organisms (e.g. Human), therefore performance on species-specific evaluation tasks are not guaranteed to transfer to other organisms. Furthermore, functional annotations in databases are rarely standardized in format, necessitating careful curation (e.g. unification of synonymous text labels requires expert knowledge). Critically, all biological sequences are related through evolu-

tion. Without carefully designed parameters, train-test leakage can frequently occur, resulting in unreliable evaluation results (Fang, 2023).

3. **Biological function takes place across diverse scales.** Single nucleotide polymorphisms can have phenotypic effects, while entire segments of genomes can be coordinated to carry out singular functions (e.g. biosynthetic gene clusters). These challenges innate to biological data have led to the lack of diverse benchmarks, resulting in independent evaluations of models on biased sets of "in-house" tasks, preventing comprehensive and objective model comparisons.

The Diverse Genomic Embedding Benchmark (DGEB) is inspired by text embedding evaluation benchmarks that have advanced the field of natural language modeling. DGEB aims to span diverse types of downstream embedding tasks, scopes of function, and taxonomic lineages. DGEB consists of 18 datasets covering 117 phyla across all three domains of life (Bacteria, Archaea and Eukarya). Similar to Massive Text Embedding Benchmark (MTEB) (Muennighoff et al., 2023), DGEB evaluates embeddings using six different embedding tasks: Classification, BiGene mining, Evolutionary Distance Similarity (EDS), Pair classification, Clustering, and Retrieval. DGEB focuses on evaluating the representations of higher-order functional and evolutionary relationships of genomic elements, and is designed to complement existing benchmarks that focus on residue-level representations ((Notin et al., 2023) (Marin et al., 2024)).

We provide DGEB as an open source software, facilitating the evaluation of custom models, and enabling the addition and revision of datasets. Biological labels for function are limited and rely on careful curation by domain experts. DGEB provides a much-needed infrastructure for allowing experts to contribute new benchmarks and revise datasets upon acquisition of new knowledge. Community driven efforts to collect and standardize diverse datasets will move the emerging interdisciplinary field of Machine Learning and Biology forward.

## 2 RELATED WORKS

### 2.1 NATURAL LANGUAGE EMBEDDING BENCHMARKS

Embedding benchmarks (e.g. SentEval (Conneau & Kiela, 2018); BEIR (Thakur et al., 2021); MTEB (Muennighoff et al., 2023)) in natural language processing (NLP) aim to evaluate how the structure of word/sentence representations match the geometric structure of their semantics. For natural language, tasks are typically either zero-shot or few-shot; examples of such tasks range from distance-based matching of translated texts to classifying tweets based on the labeled sentiment. NLP benchmarks highlight the need for holistic evaluation of models through a diverse set of tasks, as model performance can vary significantly across tasks and datasets.

### 2.2 BIOLOGICAL SEQUENCE AND LANGUAGE MODELS

Biological sequence language models are unsupervised models trained on biological sequence data such as proteins or genomic segments. Protein language models (pLMs) have been shown to encode features for protein structure prediction (Lin et al., 2023), enzyme function prediction (Yu et al., 2023) and remote homology search (Liu et al., 2024). More recently, genomic language models (gLMs) have been evaluated on classification of various genomic motifs (e.g. regulatory elements, chromatin features, splicing) (Dalla-Torre et al., 2023) and mutation fitness prediction (Nguyen et al., 2024).

### 2.3 BIOLOGICAL FUNCTION BENCHMARKS

Existing benchmarks rely mainly on two types of evaluation to measure biological function:

1. **Fitness prediction of mutations using large-scale datasets collected from deep mutational scanning (DMS) data.** DMS (Fowler & Fields, 2014) uses large-scale mutagenesis and high-throughput sequencing to model fitness landscapes of various mutations (e.g. substitutions and indels) in a single protein. ProteinGym (Notin et al., 2023) leverages diverse DMS datasets to

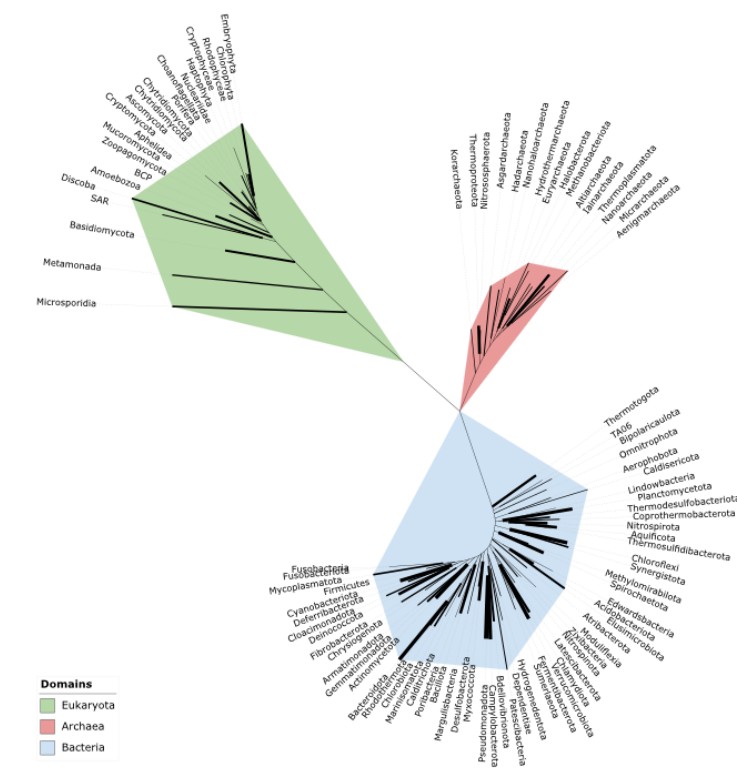

Figure 1: **Phylogenetic tree of all phyla represented in DGEB.** One representative 16S/18S sequence for each phylum represented in any DGEB dataset was obtained from SILVA (Quast et al., 2013), where available. Phylogeny was estimated using iQ-TREE 2. Widths of tree branches correspond to how well a given phylum is represented across multiple datasets.

evaluate a model's ability to predict fitness scores of mutants in either zero-shot or supervised regimes. While fitness prediction serves as a meaningful proxy for evaluating model understanding of genotype to phenotype relationships at the residue-level for a single protein, this metric cannot be used to determine how well a model can abstract evolutionary and functional relationships between non-homologous proteins.

2. **Classification of proteins on their biophysical properties.** For example, PEER (Xu et al., 2022) benchmarks protein models on various general biophysical properties, such as fluorescence, localization and solubility. These are important properties, they are too coarse in scope to evaluate whether a model has learned more granular functional information (e.g. enzymatic function, protein interaction)

Figure 2: **Overview of tasks and datasets in DGEB.** Nucleic acid (NA) and amino acid (AA) modality specific datasets are marked in purple and green respectively, and datasets that support both modalities are marked with both colors.

## 3 THE DGEB BENCHMARK

### 3.1 DESIGN CHOICES

DGEB is built on the desiderata previously outlined by NLP benchmarks, in particular, MTEB. While biological sequences and their functional labels are fundamentally different from natural language, these design choices allow for a scalable and flexible framework that can be expanded and optimized as the field matures.

a) **Diversity:** We aim to cover sequences derived from phylogenetically diverse lineages of biology (Fig. 1). Existing functional benchmarks largely consist of human or *E.coli* K-12 sequences. Data imbalance in biology is a critical problem when training biological sequence language models (Ding & Steinhardt, 2024) and prevents the models from learning features transferable to underrepresented sequences. Benchmarks that only utilize sequences from highly overrepresented sequences in the training set perpetuate this problem of data imbalance, hindering the progress towards AI-enabled characterization, discovery and design of diverse biological sequences.

b) **Simplicity:** DGEB provides a simple API that can be used with any custom model that encodes biological sequences into vectors.

c) **Extensibility:** Given the complexity of biological function, no single dataset can fully capture its diversity, and existing functional annotations must be continuously refined and expanded. DGEB supports simple extension of tasks and datasets. New or revised datasets can be uploaded to the HuggingFace Hub and new evaluation tasks can easily be added through GitHub pull requests.

d) **Reproducibility:** We version both the software and the datasets and include versioning in the results, making the benchmark results fully reproducible.

### 3.2 TASKS AND EVALUATION

DGEB consists of 18 datasets that are evaluated using one of the six task types (Fig. 2). The tasks and their evaluation schemes are described below:

**BiGene Mining** BiGene Mining is inspired by Bitext Mining tasks in NLP, where the tasks typically consist of matching translated sentences between two languages using cosine similarity. For BiGene Mining, we curated functionally analogous sequences found in two phylogenetically distant taxa (e.g. Bacteria and Archaea) or interacting pairs in sets of orthologous sequences. For each gene in the first set, the best match

in the second set is found using the cosine similarity. F1 serves as the primary metric for BacArch BiGene Mining, while recall@50 is used as the primary metric for ModBC BiGene Mining due to the difficulty of the task; accuracy and precision are also reported.

**Evolutionary Distance Similarity (EDS)**   This task evaluates how accurately models learn evolutionary relationships between sequences. We compute the correlation between pairwise embedding distances and their phylogenetic distances (sum of branch lengths connecting the two leaves of the calculated phylogenetic tree). Larger phylogenetic distance represents more evolutionary time since divergence. Pearson correlations are calculated and the top correlation score across three distance metrics (cosine, euclidean, and manhattan) is reported as the primary metric.

**Classification**   Classification tasks measure the model's ability to map from embeddings to discrete functional classes with few-shot supervision. For multiclass single-label classification, a logistic regression classifier is trained with up to 1000 iterations. For multiclass multi-label classification, a k-nearest neighbor (kNN) classifier is trained. Test performance on the test set is measured using F1 as the main metric; accuracy and average precision scores are also reported.

**Pair Classification**   Pair classification tasks evaluate model understanding of functional relationships between pairs of sequences. Inputs are pairs of sequences, where labels are binary variables denoting the existence of some particular functional relationship between the pair. Sequences are embedded and the distances between the pairs are calculated cosine similarity, dot product, euclidean distance and manhattan distance. The best binary threshold accuracy, average precision, F1, precision, and recall are calculated. The primary metric is the average precision score calculated using cosine similarity.

**Clustering**   Clustering tasks evaluate zero-shot separability of embeddings over discrete classes. Inputs are sets of sequences with labels, and a mini-batch k-means model is trained on their embeddings. The primary metric is v-measure (Rosenberg & Hirschberg, 2007).

**Retrieval**   Retrieval tasks evaluate how well a query embedding can retrieve functionally analogous sequences. Dataset consists of a corpus and queries, where the objective is to rank the embeddings in the corpus by cosine similarity to each query sequence. Correct retrieval is determined by matching functional labels. An example of a retrieval task is retrieving a bacterial homolog given an archaeal query sequence. nDCG@k, MRR@k, MAP@k, precision@k and recall@k are calculated for k=5, 10, 50. MAP@5 is used as the primary metric.

### 3.3 DATASETS

Datasets are divided into three categories: single-element, inter-element, and multi-element, where an element refers to a protein/gene or noncoding RNA. Each element can be represented in amino acid and/or nucleotide sequence modalities. Some datasets support multiple sequence modalities (AA and NA), allowing direct comparison between protein and genomic language models. Statistics for each dataset are found in Appendix B. All datasets are dereplicated at sequence identity thresholds of 70% using CD-hit (Huang et al., 2010), to remove sampling biases. For tasks requiring train and test splits, datasets are split with a maximum sequence identity of 10%. For tasks requiring multiple classes, we conduct class-balanced random sampling. Detailed preprocessing steps are found in Appendix A.

**Single Element Datasets (SE)**   For SE datasets, each genomic element (protein/gene, noncoding RNA) is individually embedded, with an associated label. SE datasets in DGEB include:

- *RNA Clustering*: rRNA, sRNA, and tRNA features predicted using RFam (Kalvari et al., 2021) genomes across diverse taxa. We cluster the sequence embeddings and assess how well they match the RNA class assignments.

- *MopB Clustering*: The dimethyl sulfoxide reductase (or MopB) family is a functionally diverse set of enzymes found across Bacteria and Archaea. Sequences are sampled from Wells et al. (2023),

where the sequence's catalytic functions are assigned using phylogenetic analysis. We assess how well the embeddings cluster with their catalytic function.

- *EC Classification*: Enzyme commission (EC) numbers are assigned to protein sequences. For each EC class, one sequence is randomly selected for testing, and four sequences from the corresponding class that have less than 10% sequence identity to each other and test sequence are selected for training.

- *Convergent Evolution Classification*: Examples of convergent evolution in proteins include enzymes that have different evolutionary history but have converged in the enzymatic reaction that they confer. We identify such convergent enzymes by curating a set of enzymes that have no sequence similarity to any of the other sequences in the train set with the same EC designation.

- *Archaeal Retrieval*: Given the corpus of bacterial protein sequences in SWISS-PROT (Bairoch & Apweiler, 2000), where the label is the corresponding text annotation, we query archaeal sequences with string match annotations in the bacterial corpus. We retrieve k nearest neighbors in bacterial corpus embedding space and look for matching labels to calculate the metrics@k.

- *Eukaryotic Retrieval*: Given the corpus of bacterial protein sequences in SWISS-PROT, we query eukaryotic sequences with string match annotations in the bacterial corpus. Metrics are calculated as above.

**Inter-element datasets (IE)** Understanding biological function relies on understanding the evolutionary and functional relationships between sequences. For IE datasets, a label is assigned for each pair of genomic embeddings. IE datasets include:

- *BacArch BiGene*: Similar to matching translated sentences between two languages, we curated functionally analogous pairs of sequences in a bacterial genome (*Escherichia coli* K-12) and an archaeal genome (*Sulfolobus acidocaldarius* DSM 639 ASM1228v1).

- *ModBC BiGene*: Identifying interacting pairs of ModB and ModC from sets of orthologs is a challenging task. ModB and ModC are interacting subunits of an ABC transporter. This dataset consists of pairs of ModB and ModC that are found to be interacting in the same genome. The goal is to correctly find the interacting ModC for each ModB given a set of orthologous ModC sequences (found in different genomes).

- *E.coli Operonic Pair Classification*: Given a pair of adjacent proteins, the label is assigned based on whether they belong to the same transcription unit in *Escherichia coli* K-12 substr. MG165.

- *Vibrio Operonic Pair Classification*: Same as *E.coli Operonic Pair Classification* except with *Vibrio cholerae* O1 biovar El Tor str. N16961.

- *Cyano Operonic Pair Classification*: Same as *Ecoli Operonic Pair Classification* except with *Synechococcus elongatus* PCC 7942.

- *FeFeHydrogenase Phylogeny*: Fe-Fe hydrogenases are complex enzymes that carry out important metabolic functions across diverse organisms. They carry out divergent and specific functions including $H_2$ production, $H_2$ sensing, $H_2$ uptake, and $CO_2$ reduction. Identifying the specific function of these hydrogenases often requires constructing a phylogenetic tree that reconstructs the evolutionary history of the catalytic, or large, subunit. This dataset includes the phylogenetic distances (sum of tree branches connecting the leaves) calculated for all pairs of Fe-Fe hydrogenase sequences.

- *RpoB Bacterial Phylogeny*: RpoB is a ribosomal protein conserved across bacteria and archaea. They are essential single-copy genes and not frequently horizontally transferred, and therefore are often used as phylogenetic marker genes. The RpoB gene is also significantly longer than the

Fe-Fe hydrogenase gene making this phylogeny distinctly different from the Fe-Fe hydrogenase phylogeny. We sample bacterial RpoB sequences utilized as markers in the GTDB (Parks et al., 2022) database, and calculate the tree to assign phylogenetic distances between pairs of RpoB sequences.

- *RpoB Archaeal Phylogeny*: Same as *RpoB Bacterial Phylogeny* but with archaeal genomes in GTDB.

- *Bacterial 16S Phylogeny*: 16S rRNA genes encode ribosomal RNA and are universal across Bacteria and Archaea. 16S rRNA is often used as a taxonomic marker gene because it rarely undergoes horizontal gene transfer and has both conserved and variable regions. Bacterial 16S rRNA sequences were downloaded from the SILVA database (Quast et al., 2013) and phylogenetic distances were calculated for each pair of sequences.

- *Archaeal 16S Phylogeny*: Same as *Bacterial 16S Phylogeny* but with archaeal sequences from SILVA.

- *Eukaryotic 18S Phylogeny*: Same as *Bacterial 16S Phylogeny* but with 18S rRNA (eukaryotic homolog of 16S rRNA) from SILVA.

**Multi-element datasets (ME)**    Many biological functions are carried out by multiple genomic elements in conjunction. DGEB supports multi-element datasets, where a label is assigned to a larger genomic sequence containing more than one genes, and whereby a single embedding is calculated either by mean-pooling across genes, or segments of genome with predefined window size. DGEB currently supports one multi-element dataset:

- *MIBiG Classification*: Minimum Information about a Biosynthetic Gene cluster (MIBiG) (Terlouw et al., 2023) is a database of biosynthetic gene clusters where a genomic segment consisting of multiple genes synthesize various classes of natural products (e.g. Polyketides, NRPS, etc). A single genomic segment can synthesize molecules that belong to multiple classes, making this a multi-label, multi-class classification task. Train and test sets are split at 80/20 using stratified random sampling.

## 4 RESULTS

### 4.1 MODELS

We focus on evaluating self-supervised models pretrained on either amino acid (AA) or nucleic acid (NA) sequences. These are "foundation models" that are not fine-tuned for specific tasks, and we evaluate how well the pre-trained embeddings capture various aspects of biological function. For AA models, we evaluate the ESM2 (Lin et al., 2023) series, ESM3 (Hayes et al., 2024) open model, the ProGen2 (Madani et al., 2023) series, and the ProtTrans (Elnaggar et al., 2022) models. For NA models, we evaluate the DNABERT-2 (Zhou et al., 2024), Nucleotide Transformer (NT) (Dalla-Torre et al., 2023) series and the Evo (Nguyen et al., 2024) models. Notably, we include both masked language models (MLM) and causal language models (CLM) in our evaluation for both data modalities. To extract sequence-level embeddings, each model's hidden layer is mean-pool across the sequence dimension, resulting in a fixed-size representation. Model information is found in Appendix C. Additionally, we provide one-hot baselines for AA and NA sequences, where the sequence is represented as one-hot vectors per position (Appendix H.

### 4.2 ANALYSIS

#### 4.2.1 LAYER PERFORMANCE

For all tasks, we test performances of mid- and last hidden layers in the model. For many of the tasks, the mid layer representation outperforms last layer representations (Fig. 3). This behavior has been noted in previous studies in both NLP (Rogers et al., 2020) and pLMs (Valeriani et al., 2023), where different

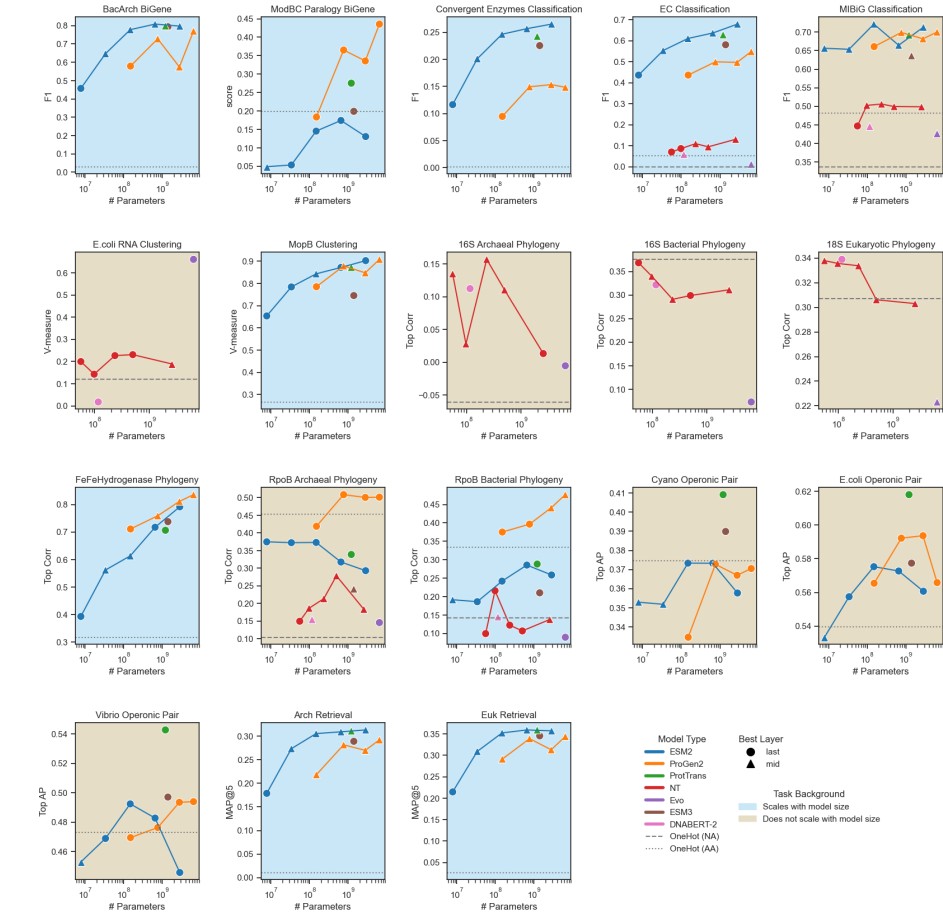

Figure 3: **Performance per task with model scaling for ESM2, ProGen2, and NT series.** Primary metric from the best scoring layer (between mid, and last) is reported for each task. Tasks where performance scales with model size for the majority of the model types are marked with a blue background. Other models plotted for reference are ProTrans, Evo, and ESM3, and DNABERT-2.

layers specialize in learning distinct semantic information. For instance, mid-layer representations for ESM2 models perform better than last layer for enzyme function classification tasks (EC Classification, Convergent Enzyme Classification) and retrieval tasks, while phylogenetic distances are better reflected in last-layer representations (RpoB phylogenies) (Appendix D). These patterns appear specific to model type. To flexibly account for this behavior, DGEB calculates model performance for both mid and last layer and reports the best score between the two.

### 4.2.2 SCALING WITH MODEL SIZE

We observe scaling with model size increase for most AA tasks, except for MIBiG classification task, RpoB archaeal phylogeny, and operonic pair tasks (Fig. 3). In general, pLMs perform poorly for predicting functions of elements that span multiple genes (e.g. biosynthetic gene clusters, operons). Additionally, while we observe improved performance with model scaling for bacterial RpoB phylogeny task, we observe no

scaling in performance for archaeal RpoB phylogeny task. This may be attributed to limitations in learning due to the significant bias against archaeal sequences in training data. Interestingly, we observe little to no evidence of improvement in performance with increasing model size for NA tasks (Fig. 3 and 4). We also test performance scaling with pre-training floating point operations (FLOPs) when the information is reported or can be derived (Appendix C). We observe scaling patterns with increasing training FLOPs (Appendix E and F) similar to those observed with increasing model size. Full results can be found in Appendix G.

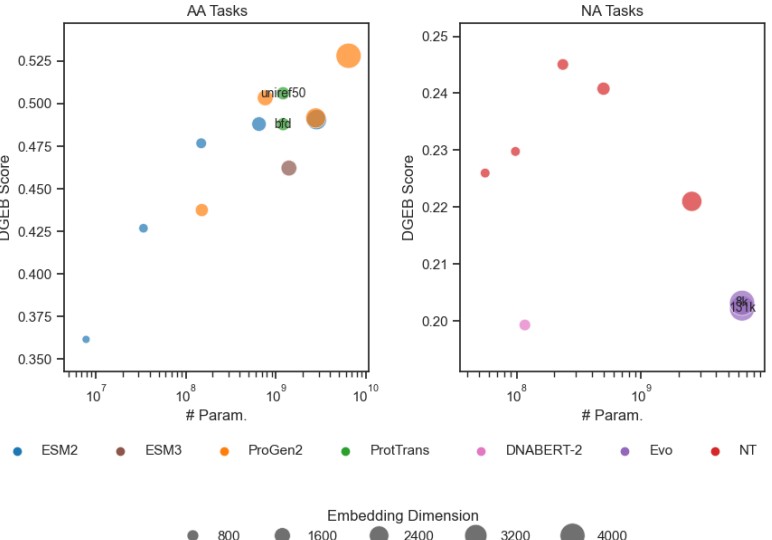

Figure 4: **Average performance across all AA and NA tasks for models benchmarked in this study.** Marker size corresponds to embedding dimension and variants of same models (e.g. evo-1-8k-base, and evo-1-131k-base) are distinguished with text labels.

### 4.2.3    DIRECT COMPARISON OF AMINO ACID AND NUCLEIC ACID MODELS

While both AA and NA sequences can be used to represent coding sequences, little work has been conducted on directly comparing the quality of NA-based model representations against AA-based model representations on the same task and data. DGEB includes four datasets that support both modalities as input for a given coding region of the sequence. For all such tasks, we find that NA sequence derived representations perform poorly in capturing biological function and evolutionary relationships of coding sequences (Fig. 5). This suggests that AA sequences are a more compute efficient input modality for learning functional information of coding sequences.

## 5    LIMITATIONS

DGEB includes multiple zero-shot tasks, as ground-truth labels for biological function are sparse and biased. These tasks rely on embedding geometry to evaluate model performance. The assumption that models capturing important features of biological function have geometry directly matching the given tasks is not guaranteed. Future research could explore methods for identifying and leveraging relevant subspaces within model embeddings. For the EDS task, we acknowledge the limitation of Euclidean embeddings for representing phylogenetic tree structures and the possibility that certain regions of the phylogeny may be of low confidence (due to the inherent uncertainty in reconstructing the ground-truth phylogeny). However, this task provides a useful starting point for comparing model performance, and will be important for evaluating

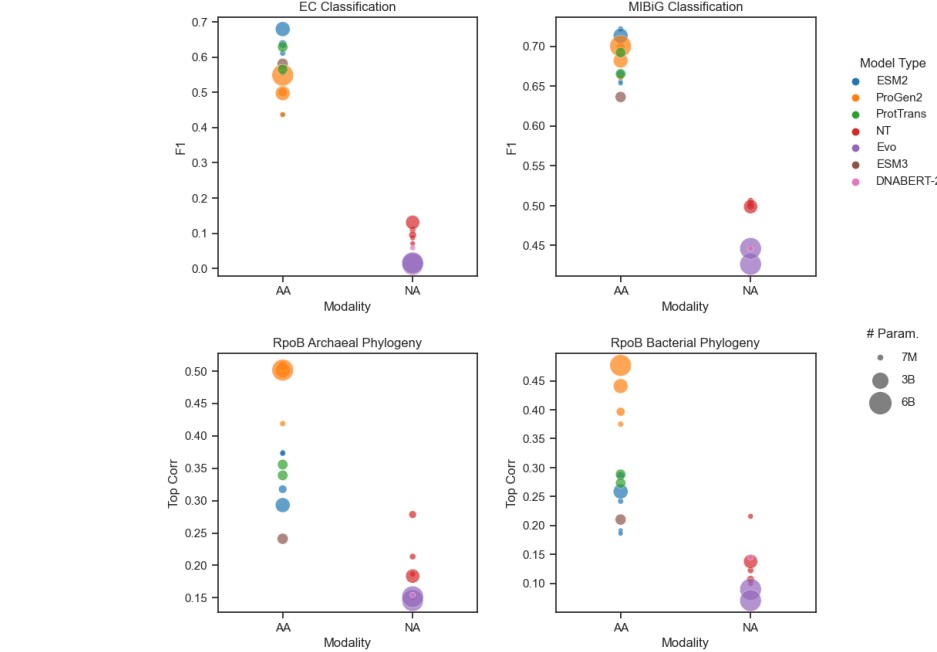

Figure 5: **Comparison of AA and NA model representations on tasks that support both modalities.** Marker color corresponds to the model type and the size corresponds to the model size.

novel hyperbolic architectures, baselined by Euclidean embedding model results. While DGEB is designed to support both NA and AA models, current suite is biased towards coding sequences with only four tasks targeting non-coding elements (16S_Arch, 16S_Bac, 16S_Euk, Ecoli_RNA), limiting ability to evaluate NA model representations of regulatory elements (e.g. promoters, transcription binding sites). Furthermore, DGEB's current evaluation suite focuses on single-element, inter-element, and multi-element scales of representations, and is designed to complement existing benchmarks that focus on residue-level representations (e.g. mutational effects (Notin et al., 2023) (Marin et al., 2024)).

## 6 CONCLUSION

We developed DGEB to assess how well learned embeddings of biological sequences capture various aspects of biological function. Our expert-curated datasets feature diverse sequences spanning all three domains and major phyla in the tree of life. We benchmarked 20 models that are trained on either AA or NA sequences. Our results demonstrate that there is no single model that performs well across all tasks. Importantly, there are many tasks where performance does not scale with model size for existing models, particularly in tasks that feature poorly represented sequences (e.g. Archaeal genes), or tasks that assess functions that require large context lengths (e.g. biosynthetic gene cluster product class classification, operon prediction). For many tasks, there is large headroom for improvement (e.g. ModBC matching, convergent enzyme classification). DGEB also supports direct comparison of models trained on AA and NA data modalities, and our results show that NA models are yet to learn important aspects of biological function. We open-source DGEB to facilitate community-driven dataset addition and revision. We hope that DGEB and the leaderboard allow transparent comparison of biological foundation models and drive the field forward.

## ETHICS STATEMENT

This study aims to advance open science by developing a open-source, reproducible benchmark for genomics. All sequences and labels are curated from public repositories. As the data originates from environmental samples, no personally identifiable information is associated with the datasets.

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

## APPENDIX A    METHODS

**ModBC BiGene Mining**    ModA and ModC sequence pairs were identified in Ovchinnikov et al. (2014) and downloaded from `https://gremlin.bakerlab.org/cplx.php?uni_a=2ONK_A&uni_b=2ONK_C`. Original sequences queried using the UniProt IDs were used for the dataset. Sequences were dereplicated at 70% sequence identity using CD-HIT (Huang et al., 2010) and only pairs where both sequences were in the dereplicated set were included in the dataset.

**BacArch BiGene Dataset**    RefSeq annotations were obtained for *E. coli* str. K-12 substr. MG1655 (GCF_000005845.2) and *Sulfolobus acidocaldarius* (GCF_000012285.1). Orthologous genes were identified as follows: 1) Genes with exactly matching annotations were identified first and added to the dataset; numerous genes with nearly identical, but not exactly matching, annotations, were also added to the dataset. 2) Genes without highly similar annotations and with matching function indicated through other databases such as UniRef, were marked as orthologs. 3) Unannotated genes in *Sulfolobus* were identified as orthologs to *E. coli* sequences through a combination of genome context information, matching HMM domains, and high structural similarity identified through a Foldseek (van Kempen et al., 2024) search between the predicted structure of *Sulfolobus* sequences against the structures available for *E. coli* MG1655; 4) For *Sulfolobus* genes with ambiguous RefSeq annotations, at least two such clues (matching UniRef annotations, genome context clues, matching HMM annotations, and Foldseek structural similarity) were required to assign an orthologous pair. 5) Genes where multiple homologs existed in both *E. coli* and *Sulfolobus* genomes were deliberately excluded.

**EC Classification Datasets**    Sequences with an assigned EC number were downloaded from UniProtKB (Breuza et al., 2016) on May 16th 2024. Only "reviewed" sequences, non-fragments and sequences with a single EC designation were included. Sequences were first dereplicated at 70% sequence identity using CD-HIT and further clustered at 10% sequence identity (`--min-seq-id 0.1`) using mmseqs cluster (Steinegger & Söding, 2017) with coverage threshold of 30% (`-c 0.3`) and minimum alignment length of 50 bp (`--min-aln-len 50`). Only EC classes with greater than five sequences after dereplication and clustering were kept. Five sequences were chosen randomly for each EC class, where one sequence was added to the test set and the remaining four were added to the train set.

**Convergent Enzymes Classification Dataset**    Raw sequences and EC labels were downloaded from UniProtKB and dereplicated at 70% sequence identity as described above in section "EC Classification." Sequences were BLASTed against every other sequence with the same EC number designation in the dereplicated set. Only one example per EC class with at least five examples in the same EC class without a significant BLASTP match (alignment length <10 and percent identity <0.1) were kept for testing. Five sequences in the corresponding EC class that have no significant BLASTP match to the test sequence were randomly chosen for training.

**MIBiG Classification Dataset**    Sequences and labels (secondary metabolite classes) were downloaded from the MIBiG server version 3.1 (`https://mibig.secondarymetabolites.org/`). Secondary metabolite class "Other" was removed from the dataset. For the AA dataset, protein sequences were extracted from the MIBiG genbank files and embedded in chunks of maximum sequence length set by `--max_seq_len` (determined by the model, e.g., 1024 for ESM2) and subsequently mean-pooled across the example. For the NA dataset, DNA sequences were extracted from the MIBiG genbank files, embedded in chunks of sequence length set by `--max_seq_len` (e.g. 8,192 for `evo-1-8k-base`, 65,536 for `evo-1-131k-base` as sequence length 131,072 did not fit into a single 80GB GPU with batch size 1) and subsequently mean-pooled to yield a single embedding per example. Examples were split into train and test sets using 80/20 ratio random sampling with stratification on the first class label.

**MopB Clustering Dataset**    Labeled MopB family sequences, displayed in the phylogenetic tree of Figure 1 in Wells et al. (2023), were obtained from their provided Supplementary Materials (`https://itol.embl.de/tree/249112161424681659917609`). Wells et al. (2023) conducted one of

the most comprehensive and up-to-date classification of MopB family enzymes to date. Sequences were first dereplicated at 70% identity using CD-HIT. Functional groups with fewer than 60 representatives were excluded from the dataset, and functional groups with greater than 100 representatives were randomly down-sampled to only include 100 representatives. Selected sequences were aligned with FAMSA (Deorowicz et al., 2016). Alignments were trimmed with trimAL (Capella-Gutiérrez et al., 2009) v1.4.rev15 with the parameter `-gt 0.1` to remove columns consisting of $\geq 90\%$ gaps. Phylogenetic trees were estimated using iQ-TREE 2 (Minh et al., 2020) with the following parameters: `-bb 1000 -m GTR+G4+F`.

**E.coli RNA Clustering**  RNA sequences in the *E. coli* str. K-12 substr. MG1655 genome (GenBank ID GCF_000005845.2) were identified by running the RFAM (Kalvari et al., 2021) family of models using the Infernal (Nawrocki, 2014) software suite. RNA groups with more than one identified representative included sRNAs, tRNAs, and rRNAs, and each sequence was classified using these three labels. In order to remove length bias in each RNA class (e.g. rRNAs are significantly longer than sRNAs), each sequence longer than 100bp was replaced by a random subsequence of length 100bp.

**RpoB Phylogenies**  RpoB sequences were obtained from the GTDB database (release 09-RS220). Bacterial RpoB sequences were identified using the TIGRFAM model TIGR02013 (rpoB_bac); Archaeal RpoB sequences were identified using the TIGRFAM model TIGR03670 (rpoB_arc) using methods described previously (Parks et al., 2018). Sequences were then dereplicated at 70% identity using CD-HIT. Sequences from phyla with fewer than 10 representatives in the GTDB were excluded. For all other phyla, 10 representative sequences were chosen; where 10 or more classes were present in each phylum, one sequence each was chosen for each of 10 random classes within the phylum in order to diversify sampled sequences, otherwise the 10 representative sequences for that phylum were chosen randomly. Nucleotide coding sequences for each chosen protein sequence were then obtained and used to construct separate phylogenies. Four phylogenies were constructed in total: Bacterial amino acid, Archaeal amino acid, Bacterial nucleotide, and Archaeal nucleotide. All alignments were performed using FAMSA. All alignments were trimmed using trimAL with parameters described above. Amino acid phylogenies were estimated using iQ-TREE 2 with the following parameters: `-bb 1000 -m LG+G4+F`. Nucleotide phylogenies were estimated using iQ-TREE 2 with the following parameters: `-bb 1000 -m GTR+G4+F`.

**FeFeHydrogenase Phylogeny**  FeFe hydrogenase catalytic subunit sequences were obtained from HydDB (Søndergaard et al., 2016) and dereplicated at 70% ID using CD-HIT. The remaining sequences were then aligned using FAMSA. Alignments were trimmed using trimAL with parameters as described above. Amino acid phylogenies were then estimated using iQ-TREE 2 with the following parameters: `-bb 1000 -m LF+G4+F`.

**16S/18S rRNA phylogenies**  16S/18S sequences were obtained from SILVA release 138_2 and dereplicated at 70% identity using CD-HIT. Sequences were then aligned with FAMSA and trimmed using trimAL with parameters as described above. Phylogenies were estimated using iQ-TREE with the following parameters: `-m GTR+G4+F+I -bb 1000` with the addition of the `+I` model parameter to accommodate the presence of invariant sites in the alignment. The phylogeny in Fig. 1 was obtained by sampling one 16S or 18S rRNA sequence from each phylum designated and constructed using the procedure described above.

**Operonic Pairs**  For transcription units information and the corresponding protein sequences were extracted from the BioCyc server (`https://biocyc.org/`) (Karp et al., 2019) for genomes *Escherichia coli* K-12 substr. MG165 *Vibrio cholerae* O1 biovar El Tor str. N16961, *Synechococcus elongatus* PCC 7942. For a given consecutive gene pair, a label was assigned (1 or 0) depending on whether or not they are found in the same transcription unit.

**Retrieval**  Protein sequences and protein name annotations were downloaded from UniProtKB on June 16 2024. Only reviewed sequences and non fragments were kept for further processing. First, the sequences were partitioned into three domain (bacterial, archaeal or eukaryotic) sets using the UniProt taxonomic designation. Second, all proteins with "UPF" or "Uncharacterized protein" in the text labels were removed.

Third, the sequence were dereplicated at 50% sequence identity with CD-HIT with additional parameters `-c 0.5, -n 2`. Finally, overlapping text annotations between bacterial and archaeal, or bacterial and eukaryotic sequence sets were identified, and only sequences that map to the overlapping text annotations were kept. For the Arch_retrieval dataset, bacterial sequences were used as corpus with archaeal sequences as query. For the Euk_retrieval dataset, bacterial sequences were used as corpus with eukaryotic sequences as query. Relevance scores for each corpus-query sequence pair were calculated using fuzzy string matching (`https://github.com/seatgeek/thefuzz`): for fuzz ratio >90 between two text annotations relevance score of 1 was assigned, otherwise, score of 0 was assigned.

## A.1 MODEL INFERENCE

For all tasks except MIBiG classification task, sequences were truncated to the model's maximum sequence length (predetermined by the model) using the flag `--max_seq_len`. For the MIBiG classification task, sequences were chunked by the model's maximum sequence length as described above.

## APPENDIX B  DATASET STATISTICS

Overview of DGEB dataset statistics. For datasets that support both modalities (amino acids (AA) and nucleic acids (NA)), the values in parenthesis refer to the statistics for NA datasets.

| Dataset | Type | Categ. | # Phyla | # Label classes | # Train | Avg. train seq length | # Test | Avg. test seq length | Modalities |
|---|---|---|---|---|---|---|---|---|---|
| BacArch | BiGene Mining | IE | 2 | 2 | - | - | 265 | 663 | AA |
| ModBC | BiGene Mining | IE | 36 | 2 | - | - | 1492 | 707 | AA |
| FeFe Hydrogenase | EDS | IE | 26 | - | - | - | 429 | 569 | AA |
| RpoB Bac | EDS | IE | 56 | - | - | - | 360 (360) | 1305 (3927) | AA, NA |
| RpoB Arch | EDS | IE | 13 | - | - | - | 170 (170) | 831 (2491) | AA, NA |
| 16S Bac | EDS | IE | 31 | - | - | - | 545 | 1686 | NA |
| 16S Arch | EDS | IE | 10 | - | - | - | 96 | 1423 | NA |
| 18S Euk | EDS | IE | 20 | - | - | - | 751 | 2117 | NA |
| Ecoli Operon | Pair Classification | IE | 1 | 2 | - | - | 4315 | 310 | AA |
| Vibrio Operon | Pair Classification | IE | 1 | 2 | - | - | 2574 | 335 | AA |
| Cyano Operon | Pair Classification | IE | 1 | 2 | - | - | 2611 | 305 | AA |
| EC | Classification | SE | 38 | 128 | 512 (512) | 541 (1901) | 128 (128) | 640 (1622) | AA, NA |
| Convergent Enzymes | Classification | SE | 51 | 400 | 2000 | 415 | 400 | 433 | AA |
| MIBIG | Classification | ME | 15 | 6 | 29992 (1763) | 647 (41178) | 7213 (441) | 638 (38206) | AA, NA |
| MopB | Clustering | SE | 46 | 13 | - | - | 1300 | 817 | AA |
| Ecoli RNA | Clustering | SE | 1 | 3 | - | - | 161 | 83 | NA |
| Arch | Retrieval | SE | 52 | - | 9229 | 344 | 2343 | 332 | AA |
| Euk | Retrieval | SE | 44 | - | 3202 | 353 | 311 | 367 | AA |

# APPENDIX C    MODEL INFORMATION AND STATISTICS

Models evaluated with DGEB are detailed with the number of parameters, number of hidden layers, and embedding dimensions. Pretraining FLOPs are estimated in (Chen et al., 2024) or from the model's original papers when available. FLOP values with an asterisk are estimated using the formula $C = 6ND$ from (Kaplan et al., 2020), where $C$ is the total pretraining flops, $N$ is the model size, and $D$ is the number of pretraining tokens.

| Model type | Model Name | Modeling Objective | Training Data | Num Params | Num Layers | Emb. Dim. | Modality | Pretrain FLOPs |
|---|---|---|---|---|---|---|---|---|
| ESM2 | esm2_t6_8M_UR50D | MLM | UniRef50/D | 8M | 6 | 320 | AA | 4.8E+19* |
| ESM2 | esm2_t12_35M_UR50D | MLM | UniRef50/D | 35M | 12 | 480 | AA | 2.1E+20* |
| ESM2 | esm2_t30_150M_UR50D | MLM | UniRef50/D | 150M | 30 | 640 | AA | 1.1E+21 |
| ESM2 | esm2_t33_650M_UR50D | MLM | UniRef50/D | 650M | 33 | 1280 | AA | 4.4E+21 |
| ESM2 | esm2_t36_3B_UR50D | MLM | UniRef5/0D | 3B | 36 | 2560 | AA | 1.9E+22 |
| ESM3 | esm3_sm_open_v1 | MLM | UniRef, MGnify; JGI (Hayes et al. 2024) | 1.4B | 48 | 1536 | AA | 6.72E+20 |
| ProGen | progen2-small | CLM | UniProtKB | 150M | 12 | 1024 | AA | 1.8E+20 |
| ProGen | progen2-medium | CLM | UniProtKB | 765M | 27 | 1536 | AA | 8.9E+20 |
| ProGen | progen2-large | CLM | UniProtKB | 2.7B | 32 | 2560 | AA | 3.4E+21 |
| ProGen | progen2-xlarge | CLM | UniProtKB | 6.4B | 32 | 4096 | AA | 1.4E+22 |
| ProTrans | prot_t5_xl_uniref50 | MLM | UniRef50 | 1.2B | 24 | 1024 | AA | - |
| ProTrans | prot_t5_xl_bfd | MLM | BFD (Steinegger and Söding 2018) | 1.2B | 24 | 1024 | AA | 1.7E+22 |
| NT | nt-v2-50m-multi-species | MLM | Multispecies (NCBI) (Dalla-Torre et al. 2023) | 55M | 12 | 512 | NA | 9.0E+19* |
| NT | nt-v2-100m-multi-species | MLM | Multispecies (NCBI) | 98M | 22 | 512 | NA | 1.76E+20* |
| NT | nt-v2-250m-multi-species | MLM | Multispecies (NCBI) | 235M | 24 | 768 | NA | 1.13E+21* |
| NT | nt-v2-500m-multi-species | MLM | Multispecies (NCBI) | 498M | 29 | 1024 | NA | 2.69E+21* |
| NT | nt-2.5b-multi-species | MLM | Multispecies (NCBI) | 2.5B | 32 | 2560 | NA | 4.5E+21* |
| Evo | evo-1-8k-base | CLM | OpenGenome (Nguyen et al. 2024) | 6.5B | 32 | 4096 | NA | - |
| Evo | evo-1-131k-base | CLM | OpenGenome | 6.5B | 32 | 4096 | NA | 2E+22 |
| DNABERT | DNABERT2 | MLM | Multispecies | 117M | 12 | 768 | NA | 2.3E+20 |

# APPENDIX D    COMPARISON OF MID LAYER AND LAST LAYER PERFORMANCE FOR ESM2 SERIES MODELS.

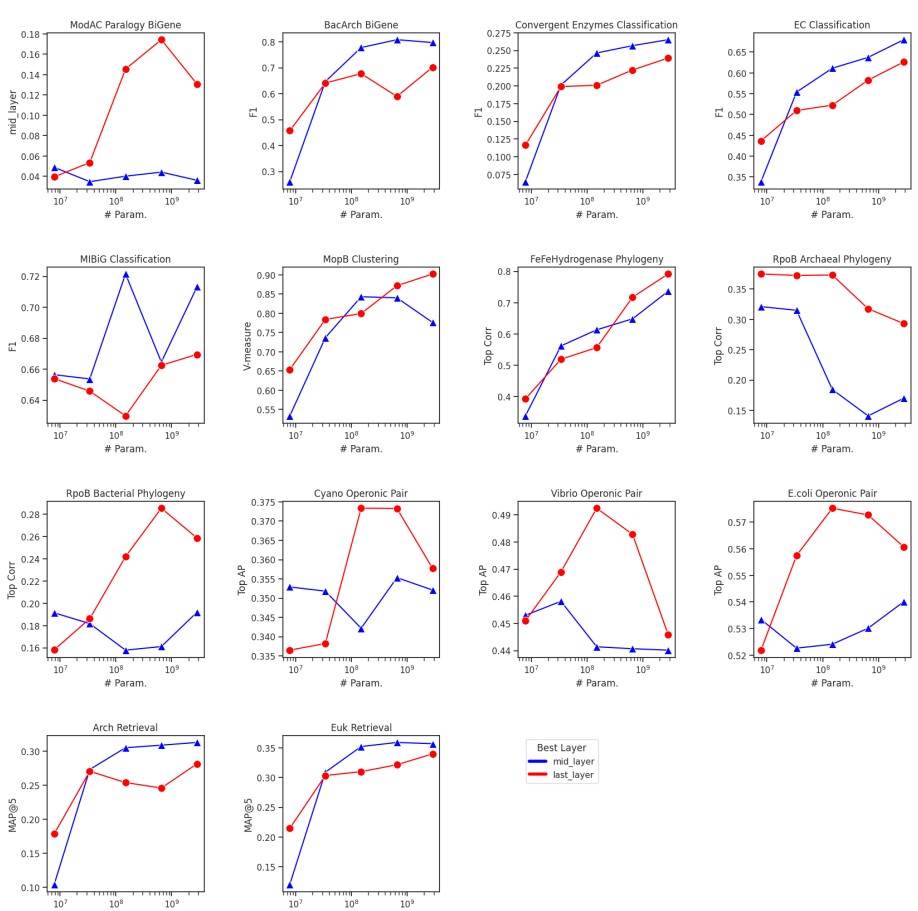

# APPENDIX E    PER-TASK PERFORMANCE SCALING WITH PRE-TRAINING FLOPS.

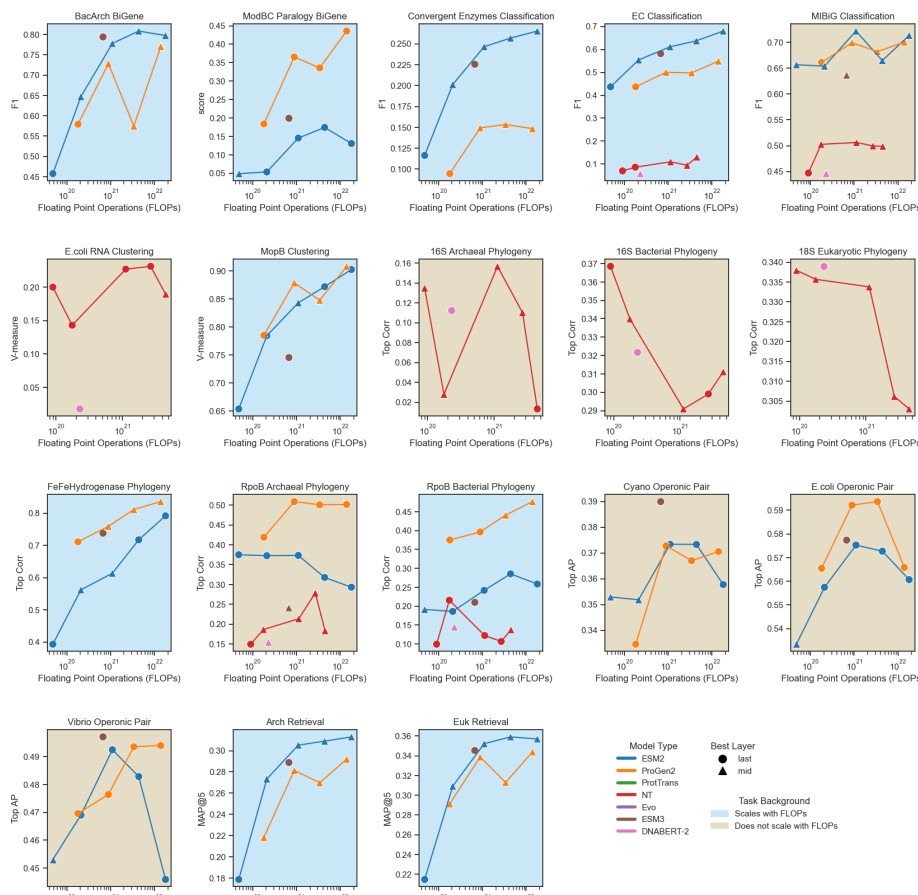

# APPENDIX F    AGGREGATED DGEB SCORE RELATIVE TO PRE-TRAINING FLOPS.

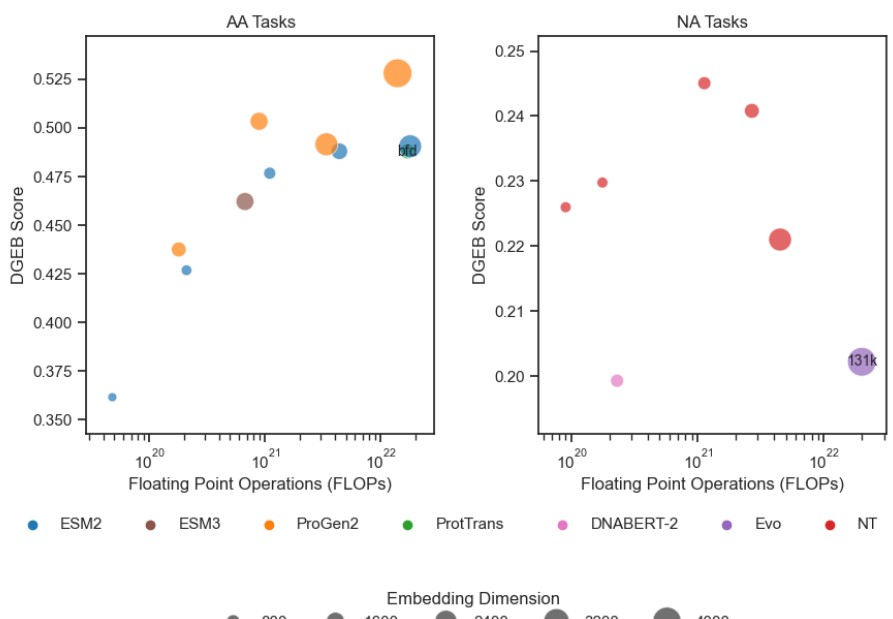

# APPENDIX G   MODEL PERFORMANCE PER TASK

24

| Task Type | Task | AA models | | | | | | | | | | | | NA models | | | | | | | |
| | | ESM2 | | | | | ESM3 | Progen | | | | ProtTrans | | Nucleotide Transformer | | | | | Evo | | DNABERT |
| | | esm2 t6_8M UR50D | esm2 t12_35M UR50D | esm2 t30_150M UR50D | esm2 t33_650M UR50D | esm2 t36_3B UR50D | esm3 3B U50D | progen2-small | progen2-medium | progen2-large | progen2-xlarge | prot_t5_xl uniref50 | prot_t5_xl bfd | NT v2-50m Multispecies | NT v2-100m Multispecies | NT v2-250m Multispecies | NT v2-500m Multispecies | NT 2.5b Multispecies | evo-1 8k-base | evo-1 131k-base | DNABERT2 117M |
|---|---|---|---|---|---|---|---|---|---|---|---|---|---|---|---|---|---|---|---|---|---|
| BiGene Mining | ModBC BiGene | 0.049 | 0.054 | 0.145 | 0.174 | 0.131 | 0.199 | 0.184 | 0.365 | 0.336 | **0.436** | 0.275 | 0.273 | n/a | n/a | n/a | n/a | n/a | n/a | n/a | n/a |
| | BacArch BiGene | 0.457 | 0.647 | 0.778 | **0.808** | 0.797 | 0.794 | 0.579 | 0.728 | 0.575 | 0.770 | 0.799 | 0.782 | n/a | n/a | n/a | n/a | n/a | n/a | n/a | n/a |
| Classification | EC Classification | 0.437 | 0.554 | 0.611 | 0.637 | **0.680** | 0.581 | 0.437 | 0.500 | 0.497 | 0.549 | 0.629 | 0.565 | 0.070 | 0.086 | 0.110 | 0.095 | 0.131 | 0.012 | 0.016 | 0.086 |
| | MIBIG Classification | 0.656 | 0.654 | **0.722** | 0.665 | 0.713 | 0.636 | 0.661 | 0.699 | 0.682 | 0.700 | 0.692 | 0.665 | 0.447 | 0.503 | 0.506 | 0.500 | 0.499 | 0.426 | 0.446 | 0.446 |
| | Convergent Enzymes Classification | 0.116 | 0.201 | 0.246 | 0.257 | 0.265 | 0.225 | 0.095 | 0.149 | 0.153 | 0.148 | 0.243 | 0.227 | n/a | n/a | n/a | n/a | n/a | n/a | n/a | n/a |
| Clustering | MopB Clustering | 0.654 | 0.784 | 0.843 | 0.872 | 0.902 | 0.745 | 0.785 | 0.879 | 0.848 | **0.908** | 0.872 | 0.828 | n/a | n/a | n/a | n/a | n/a | n/a | n/a | n/a |
| | E. coli RNA Clustering | n/a | n/a | n/a | n/a | n/a | n/a | n/a | n/a | n/a | n/a | n/a | n/a | 0.200 | 0.143 | 0.227 | 0.231 | 0.190 | 0.660 | **0.681** | 0.018 |
| EDS | FeFeHydrogenase Phylogeny | 0.393 | 0.562 | 0.614 | 0.717 | 0.792 | 0.738 | 0.711 | 0.759 | 0.811 | **0.839** | 0.707 | 0.624 | n/a | n/a | n/a | n/a | n/a | n/a | n/a | n/a |
| | 16S Bacterial Phylogeny | n/a | n/a | n/a | n/a | n/a | n/a | n/a | n/a | n/a | n/a | n/a | n/a | **0.368** | 0.340 | 0.291 | 0.299 | 0.311 | 0.073 | 0.073 | 0.322 |
| | 16S Archaeal Phylogeny | n/a | n/a | n/a | n/a | n/a | n/a | n/a | n/a | n/a | n/a | n/a | n/a | 0.135 | 0.028 | **0.157** | 0.110 | 0.013 | -0.005 | 0.019 | 0.112 |
| | 18S Eukaryotic Phylogeny | n/a | n/a | n/a | n/a | n/a | n/a | n/a | n/a | n/a | n/a | n/a | n/a | **0.338** | 0.336 | 0.334 | 0.306 | 0.303 | 0.223 | 0.161 | 0.339 |
| | RpoB Archaeal Phylogeny | 0.375 | 0.372 | 0.373 | 0.318 | 0.293 | 0.241 | 0.419 | **0.509** | 0.501 | 0.501 | 0.339 | 0.356 | 0.150 | 0.187 | 0.214 | 0.279 | 0.184 | 0.146 | 0.152 | 0.154 |
| | RpoB Bacterial Phylogeny | 0.191 | 0.186 | 0.242 | 0.286 | 0.259 | 0.210 | 0.375 | 0.397 | 0.441 | **0.477** | 0.288 | 0.273 | 0.100 | 0.216 | 0.123 | 0.107 | 0.138 | 0.090 | 0.070 | 0.145 |
| Pair Classification | E. coli Operonic Pair | 0.533 | 0.557 | 0.575 | 0.573 | 0.561 | 0.577 | 0.565 | 0.592 | 0.594 | 0.566 | 0.618 | **0.626** | n/a | n/a | n/a | n/a | n/a | n/a | n/a | n/a |
| | Cyano Operonic Pair | 0.353 | 0.352 | 0.373 | 0.373 | 0.358 | 0.390 | 0.335 | 0.373 | 0.367 | 0.371 | **0.409** | 0.407 | n/a | n/a | n/a | n/a | n/a | n/a | n/a | n/a |
| | Vibrio Operonic Pair | 0.453 | 0.469 | 0.492 | 0.483 | 0.446 | 0.497 | 0.470 | 0.476 | 0.494 | 0.494 | **0.543** | 0.541 | n/a | n/a | n/a | n/a | n/a | n/a | n/a | n/a |
| Retrieval | Euk Retrieval | 0.215 | 0.309 | 0.352 | **0.359** | 0.357 | 0.345 | 0.291 | 0.339 | 0.313 | 0.344 | 0.359 | 0.355 | n/a | n/a | n/a | n/a | n/a | n/a | n/a | n/a |
| | Arch Retrieval | 0.179 | 0.273 | 0.305 | 0.309 | **0.313** | 0.289 | 0.218 | 0.281 | 0.270 | 0.292 | 0.311 | 0.306 | n/a | n/a | n/a | n/a | n/a | n/a | n/a | n/a |

## APPENDIX H    ONE-HOT BASELINE DETAILS

We introduce a one-hot vector representation of biological sequences as a baseline method to compare model performance. This baseline represents each residue or nucleic acid as a one-hot vector. The one-hot representation is mean-pooled across the sequence dimension, like all evaluated models (Section 4.1).

After mean-pooling, the one-hot baseline results in a representation equivalent to amino-acid composition for AA tasks and nucleic-acid composition for NA tasks. Previous work has shown that amino-acid composition is highly predictive of biological tasks such as transmembrane $\beta$-barrel protein identification (Garrow et al., 2005) and microbial growth conditions (Barnum et al., 2024). The baseline representation is evaluated in the same way as model embeddings, such as logistic regression for single-label classification tasks.