# OpenReview forum: "Diverse Genomic Embedding Benchmark for Functional Evaluation Across the Tree of Life"
_ICLR.cc/2025/Conference — Submitted to ICLR 2025_

### Official Review · Reviewer_Fdye · 2024-10-22

**Soundness:** 2
**Presentation:** 3
**Contribution:** 3
**Rating:** 6
**Confidence:** 4

**Summary:**

This paper proposes a new benchmarking suite to evaluate the ability of protein and genomic language models to accurately represent cellular function. The authors claim that most existing benchmarks focus on structural properties instead of function, and highlight several challenges with evaluating accurate functional representation. The authors then define six categories of tasks in their benchmarking suite, and aggregate 18 datasets falling into these categories that capture aspects of cellular function. Tasks involve predicting properties of individual sequences as well as pair-wise or multi-sequence interactions. The authors then benchmark several protein and genomic foundation models on the previous tasks, and highlight comparative performance as well as the presence of scaling law behaviour.

NOTE: Raised score from 5->6 in response to rebuttals.

**Strengths:**

- Selection of datasets capture meaningful aspects of protein function and phylogeny, and cover a wide range of prediction tasks. Organization of datasets into hierarchy based on prediction task and relationship conceivably would help extension of the dataset by integration of future datasets.
- Authors highlight the importance of controlling data leakage that might occur through sequence homology, and construct train / test splits that avoid this using several sequence identity thresholds. This contributes towards the rigour of models that would be benchmarked on DGEM, and raises the current standard in the genomics space. An additional contribution could be to include code to regenerate splits based on a random seed; this would be useful for generating CIs.
- Benchmarking of existing foundation models on the selected tasks reveal interesting empirical evidence on the presence of scaling laws and performance gaps between existing AA and NA models.

**Weaknesses:**

- Although the selected tasks / datasets do represent aspects of biological function, I am concerned that the overall claims of the paper are too large in its current state. Many aspects of biological function that could be represented by a foundation model embedding are not tested for in this dataset, with most of the proposed tasks evaluating enzymatic function or evolutionary similarity between proteins. In particular, attention to tasks specific to NA sequences that could be used to evaluate gLMs are relatively lacking in comparison to protein tasks. For example, the classification of sequence into annotation classes as defined by the Gene Ontology resource is commonly used, or classification of RNA splice isoform function or tissue-specific expression could be a task related to function that evaluates gLMs. I believe the paper should either seek to restrict the breadth of their claims, or incorporate other commonly used measures of biological function into their dataset.
- As a benchmarking paper, I believe the authors are well positioned to further increase the relevance of their work by inclusion of baselines that are not derived from a foundation model. It has been shown in other biological foundation models that pre-trained embeddings often perform worse than using a linear model on a set of naively engineered features, and addition of results similar to this could contextualize the performance of the benchmarked foundation models. Similarly, it would also be extremely relevant if performance of an ab-initio supervised model on the provided tasks are shown. This would again contextualize the actual benefit gained from the often costly foundation model pre-training.
- The contribution of the paper could be strengthened by benchmarking more gLM models. While the overall conclusions of the AA vs NA experiment are believable given the known weaknesses of applying NLP objectives on genomic data, including a more diverse set of benchmarked gLMs could strengthen the claim.

**Questions:**

- An anonymized version of the code/data repository does not seem available, meaning it is difficult to assess some core claims regarding DGEB's implementation (extensibility, reproducibility, etc.). Is there any way to provide a snapshot of the implementation via OpenReview or some other anonymized github repository?

- The colour scheme for Figure 3's Model Type is not the most colourblind friendly, especially the colours for NT / ProtTrans / ESM3. I would appreciate it if the figure could be adjusted for easier reading. As a suggestion, maybe the gLMs could be outlined in black as an additional dimension for visual clarity?

---

> ### Author Response · Authors · 2024-11-22
> **Response to reviewer Fdye**
>
> Thank you for your thoughtful review and suggestions. We appreciate you deeming DGEB to "raises the current standard in the genomics space" and our findings to "reveal interesting empirical evidence on the presence of scaling laws and performance gaps between existing AA and NA models"
>
> > I am concerned that the overall claims of the paper are too large in its current state.
>
> Thank you for raising this. We acknowledge that our DGEB benchmarking suite is not comprehensive of all aspects of biological function. We have included a limitations section to highlight areas where DGEB currently does not support. In addition, in the introduction we add this following sentence highlighting where DGEB fits into the existing Benchmark landscape. "DGEB focuses on evaluating the representations of higher-order functional and evolutionary relationships of genomic elements, and is designed to complement existing benchmarks that focus on residue-level representations (Notin et al., (2023), Marin et al., (2024))."
>
> Please see the added limitation section pasted below: "DGEB includes multiple zero-shot tasks, as ground-truth labels for biological function are sparse and biased. These tasks rely on embedding geometry to evaluate model performance. The assumption that models
> capturing important features of biological function have geometry directly matching the given tasks is not guaranteed. Future research could explore methods for identifying and leveraging relevant subspaces within model embeddings. For the EDS task, we acknowledge the limitation of Euclidean embeddings for representing phylogenetic tree structures and the possibility that certain regions of the phylogeny may be of low confidence (due to the inherent uncertainty in reconstructing the ground-truth phylogeny). However, this
> task provides a useful starting point for comparing model performance, and will be important for evaluating novel hyperbolic architectures, baselined by Euclidean embedding model results. While DGEB is designed to support both NA and AA models, current suite is biased towards coding sequences with only four tasks targeting non-coding elements (16S Arch, 16S Bac, 16S Euk, Ecoli RNA), limiting ability to evaluate NA model representations of regulatory elements (e.g. promoters, transcription binding sites). Furthermore,
> DGEB’s current evaluation suite focuses on single-element, inter-element, and multi-element scales of representations, and is designed to complement existing benchmarks that focus on residue-level representations (e.g. mutational effects Notin et al. (2023) Marin et al. (2024))."
>
> >  I believe the authors are well positioned to further increase the relevance of their work by inclusion of baselines that are not derived from a foundation model.
>
> Thank you for this suggestion. We have added a one-hot baseline, where the sequence is represented as one-hot vector per residue, and mean-pooled across the sequence dimension like all benchmarked models. The baseline results are shown in Figure 3 as dotted lines.
>
>
> > The contribution of the paper could be strengthened by benchmarking more gLM models.
>
> We have added the DNABERT-2 model as an additional gLM model. Our benchmark now includes all commonly used gLM models referenced in [1] (Evo is a newer version of HyenaDna from the same author, trained on diverse genomic data, not just human).
>
> > Is there any way to provide a snapshot of the implementation via OpenReview or some other anonymized github repository?
>
> We have added the anonymized code as a zip file in the OpenReview supplementary data.
>
> > The colour scheme for Figure 3's Model Type is not the most colourblind friendly.
>
> Thank you for bringing this to our attention. We will update the colors in the camera-ready revision.
>
> [1] https://www.biorxiv.org/content/10.1101/2024.08.16.608288v1

---

> ### Comment · Reviewer_Fdye · 2024-11-22
>
> Thank you for the detailed responses. I believe the changes have improved the quality of the paper and I have raised my score. Overall, as the other reviewers also point out, adding rigour in the form of homology splitting etc. is an important benchmarking requirement that this paper could help establish in the genomics community.
>
> I would like to re-iterate the sentiment shared in the original review and by other reviewers that extending this benchmark to evaluate models other than pre-trained foundation models is highly relevant to the larger community. While I acknowledge that papers have a finite scope, it is important to understand whether these pre-trained foundation models confer any advantage in the selected tasks compared to supervised deep learning on the labels from scratch. This point is reinforced by the surprising fact that NA models underperform a OHE in some tasks. Perhaps for the camera ready, the authors could add a supervised training benchmark using a simple sequence model (e.g. dilated CNN). Alternatively, I have anecdotally found that compressing the sequence using the Mamba architecture without any training at all (i.e. leveraging the structured state space initialization) is a strong naive baseline on genomic tasks.
>
> Unfortunately, I must point out that the anonymous code is decidedly not anonymous and may violate double blind. I am unsure of how to resolve this and will defer to the AC.

---

> > ### Author Response · Authors · 2024-11-22
> > **Response to Reviewer Fdye**
> >
> > > I must point out that the anonymous code is decidedly not anonymous
> >
> > Thank you for bringing this to our attention. We made best efforts to remove all identification from the codebase. Unfortunately we missed one instance and this has been corrected.

---

### Official Review · Reviewer_idyz · 2024-10-23

**Soundness:** 3
**Presentation:** 2
**Contribution:** 2
**Rating:** 5
**Confidence:** 4

**Summary:**

This paper presents the Diverse Genomic Embedding Benchmark (DGEB), which is focused on evaluating protein and genomic language models on a set of six embedding tasks (inspired from MTEB) across 18 datasets.

**Strengths:**

- The authors introduce tasks that investigate a diverse range of species, many of which seem underexplored in the existing protein language model (pLM) and genomic language model (gLM) literature.
- The tasks appear to be biologically and evolutionarily relevant.

**Weaknesses:**

- This paper deals with two different classes of models - AA models and NA models, which generally have very different underlying assumptions and draw from separate corresponding bodies of work. This dual focus weakens the depth of analysis, as attempting to tackle both classes leads to a more superficial investigation. A more focused approach, concentrating on a single class—such as AA models—would result in a stronger, more detailed exploration, particularly since most tasks appear to be better suited to this domain.
- The dataset used in the paper appears ill-suited for NA models, as the tasks primarily target protein sequences rather than regulatory or non-coding regions of the genome. This limits the applicability of the dataset to AA models, reducing the relevance and transferability for broader genomic applications.
- The DGEB datasets are relatively small and designed primarily for assessing zero-shot performance, where no training data is provided. This narrow scope restricts the benchmark’s utility to specific use cases, such as evaluating pre-trained models, rather than facilitating a comprehensive analysis of model performance across varied settings.
- The benchmarking of models could be expanded to include a wider range of architectures. A deeper exploration of model performance would enhance the analysis, providing more meaningful comparisons and insights into the strengths and limitations of different approaches.
- For each of the datasets, it would be helpful to provide more detailed explanations of the intended learning objectives and their relevance. This additional context would clarify the specific goals of the tasks and highlight their importance within the broader scope of the study.

**Questions:**

- What does the performance on these tasks/datasets reveal about the models tested? As presented in Appendix G, it seems that no single model performs best across all tasks. Therefore, a way to interpret these results seems necessary for this benchmark.
- Given the large number of protein language models, what is the justification for benchmarking the models that were picked (over others such as AlphaFold, pLM-BLAST, etc)?
- What measures are being taken to prevent data leakage in the tasks that have training data?
- For the convergent evolution classification dataset, how do you determine/enforce enzymes having different evolutionary histories?
- Given that phylogenies can only be inferred, how robust is the phylogeny construction to perturbations in parameters/underlying construction algorithm?

---

> ### Author Response · Authors · 2024-11-22
> **Response to Reviewer idyz Part 1**
>
> Thank you for taking the time to review our work. We are glad that inclusion of "diverse range of species" to be an important and "biologically and evolutionarily relevant" to benchmarking gLMs and pLMs.
>
> >This dual focus weakens the depth of analysis, as attempting to tackle both classes leads to a more superficial investigation. A more focused approach, concentrating on a single class—such as AA models—would result in a stronger, more detailed exploration, particularly since most tasks appear to be better suited to this domain.
>
> We agree that AA models and NA models traditionally draw from separate bodies of work. However, AA and NA models both fundamentally aim to represent the biological code, and more recently, the field has observed the convergence of two model types. For instance, Evo (Ngyuen et al 2024) and MegaDNA (Shao and Yan, 2024) claim that NA models can be used to generate protein coding regions and learn meaningful representations of protein sequences. Whether NA models are as efficient as AA models at learning representations of protein coding genes is an important question for the field, which could not be answered because no existing benchmark supported direct comparison of NA and AA models. We agree that the current version of DGEB features more AA tasks, we hope to expand upon NA tasks in our future DGEB releases. We believe the direct comparison of AA and NA models on representations of protein coding genes is an important contribution to the field, and could help guide machine learning practitioners in their design choices on the modality with which one models biology.
>
> > The dataset used in the paper appears ill-suited for NA models, as the tasks primarily target protein sequences rather than regulatory or non-coding regions of the genome.
>
> The two NA foundation models benchmarked here (Evo and NT) have been proposed to model protein coding genes as well as regulatory regions of the genome. For example, Evo designs Cas proteins and Transposases, and is evaluated on protein fitness and gene essentiality tasks. Recent follow-up work (Boshar et al 2024) claims that NT can be used for protein downstream tasks. With the growing efforts and motivation to model the entire genomic regions including protein coding regions with DNA sequences only (other examples: megaDNA, plasmidGPT), we believe that it is important to evaluate NA models performance on protein tasks and compare directly with AA models. We however agree that more non-coding region tasks should be included (currently limited to three rRNA EDS tasks and one 16S rRNA clustering task). We hope to include more regulatory element classification tasks in future releases of DGEB and we include this as a current limitation with the inclusion of the following sentence: " While DGEB is designed to support both NA and AA models, current suite is biased towards coding sequences with only four tasks targeting non-coding elements (16S_Arch, 16S_Bac, 16S_Euk, Ecoli_RNA), limiting ability to evaluate NA model representations of regulatory elements (e.g. promoters, transcription binding sites)."
>
> > The DGEB datasets are relatively small and designed primarily for assessing zero-shot performance, where no training data is provided.
>
> We deliberately prioritize zero-shot tasks in DGEB to address the problem of biased, sparse and often incorrect labels in biological evaluation datasets. Relying heavily on supervised benchmarks in biology can lead to biased evaluation that is not desirable for a foundation model.
>
> > This narrow scope restricts the benchmark’s utility to specific use cases, such as evaluating pre-trained models, rather than facilitating a comprehensive analysis of model performance across varied settings.
>
> DGEB is designed as a benchmark for pre-trained foundation models, not fine-tuned models. We designed DGEB to address the gap, where no diverse suite of benchmarks exists to evaluate new types of foundation models. We hope that thorough benchmarking of foundation models across diverse tasks can guide fine-tuning efforts.
>
> > The benchmarking of models could be expanded to include a wider range of architectures.
>
> We expanded the benchmarked models to include DNABERT2 in this revision. With this addition, we believe we have included most widely used and performant foundation models.

---

> ### Author Response · Authors · 2024-11-22
> **Response to Reviewer idyz Part 2**
>
> > For each of the datasets, it would be helpful to provide more detailed explanations of the intended learning objectives and their relevance.
>
> We include the biological relevance of each dataset in section 3.3. For example: "BacArch BiGene: Similar to matching translated sentences between two languages, we curated functionally analogous pairs of sequences in a bacterial genome (Escherichia coli K-12) and an archaeal genome (Sulfolobus acidocaldarius DSM 639 ASM1228v1). ModBC BiGene: Identifying interacting pairs of ModB and ModC from sets of orthologs is a challenging task. ModB and ModC are interacting subunits of an ABC transporter. This dataset consists of pairs of ModB and ModC that are found to be interacting in the same genome. The goal is to correctly find the interacting ModC for each ModB given a set of orthologous ModC sequences (found in different genomes)."
>
> > What does the performance on these tasks/datasets reveal about the models tested?
>
> The reviewer is correct in that "[o]ur results demonstrate that there is no single model that performs well across all tasks." This is similar to the conclusion drawn from analogous studies in text embedding benchmarks (e.g.  MTEB (Massive Text Embedding Benchmark) [1]) where no single foundation model performs well across all tasks. DGEB, similar to MTEB, allows the ML practitioners to choose the most appropriate model for fine-tuning depending on the specific downstream task and modeling objectives. (e.g. for search capabilities, it would be more appropriate to fine-tune a foundation that performs well in retrieval tasks)
>
> > Given the large number of protein language models, what is the justification for benchmarking the models that were picked (over
> others such as AlphaFold, pLM-BLAST, etc)?
>
> DGEB is an embedding benchmark that evaluates unsupervised biological sequence foundation models. AlphaFold is a structure prediction model that takes in an MSA and is supervised on structures. pLM-BLAST is a tool for remote homology detection built on ProtTrans (benchmarked in our study).
>
> > What measures are being taken to prevent data leakage in the tasks that have training data?
>
> We conduct extensive sequence similarity based dereplication to prevent data leakage (e.g. "For tasks requiring train and test splits, datasets are split with a maximum sequence identity of 10%.")
>
> > For the convergent evolution classification dataset, how do you determine/enforce enzymes having different evolutionary histories?
>
> Convergent evolution dataset consists of enzymes that belong to the same enzyme class (identical EC number) but have less than 10% sequence identity. (The typical threshold used for homology (shared evolutionary history) is 30% sequence similarity, see [2])
>
> > Given that phylogenies can only be inferred, how robust is the phylogeny construction to perturbations in parameters/underlying construction algorithm?
>
> It is true that there is no absolute "ground truth" for phylogenies. We use IQ-tree, which tests multiple substitution models and parameters, for determining the most likely phylogeny values. Therefore, the final tree is robust to changes in parameters in the phylogeny algorithm. However, we acknowledge that certain regions of the phylogeny can be of low confidence in the limitation section: "For the EDS task, we acknowledge [...] the possibility that certain regions of the phylogeny may be of low confidence (due to the inherent uncertainty in reconstructing the ground truth phylogeny)."
>
> [1] https://arxiv.org/abs/2210.07316
>
> [2] https://academic.oup.com/peds/article-abstract/12/2/85/1550637?redirectedFrom=fulltext

---

> > ### Comment · Reviewer_idyz · 2024-11-25
> >
> > Thank you to the authors for your response. While I am grateful for the clarification provided on some points, there are still a number of concerns I have.
> >
> > First, I remain unconvinced that the dataset is well-suited for NA models, especially given that I have to appraise DGEB in its current state and not based on potential future improvements.
> >
> > > We expanded the benchmarked models to include DNABERT2 in this revision. With this addition, we believe we have included most widely used and performant foundation models.
> >
> > To clarify, in asking for a “wider range of architectures”, it would be useful to include the foundation models that are not transformer based, such as Caduceus (a state-space model)  [1], a point which was also raised by Reviewer Fdye. Given that the benchmark can only be used to compare pretrained foundation models, I would hope that the authors be comprehensive in their benchmarking of any relevant pretrained foundation models.
> >
> > [1] https://caduceus-dna.github.io/
> >
> > Next, since it appears to be the case that no single model performs well across all tasks, the authors should more explicitly guide ML practitioners on the use cases of the tasks. Otherwise, it seems difficult to apply this benchmark as a meaningful metric.  I agree with Reviewer tQZv in that extra care needs to be put into providing comprehensive explanations for each benchmark. Since the task designs are borrowed from MTEB, an altogether different domain, it would be useful to build intuition on what each task reveals about sequence relationships. As an illustrative example - how do the classification tasks differ from clustering in terms of what they tell you about sequences? To me, it seems that they fundamentally reveal the same thing - the relationship between sequences within class labels. Therefore, it becomes difficult for me to prioritize models where one model is better at clustering but another model is better at classification (e.g. NT vs. Evo, ProGen2 vs. ESM2).
> >
> > Given these concerns, I keep my score as is for the current iteration of this paper.

---

> > > ### Author Response · Authors · 2024-12-02
> > >
> > > Thank you for the detailed response, and we would like to follow up with the reviewer's three concerns:
> > >
> > > > I remain unconvinced that the dataset is well-suited for NA models
> > >
> > > The tasks which support both AA and NA modalities allow direct comparison between pLMs and gLMs by providing the models with the same sequences, but in different modalities. We find that pLMs have significantly better performance on these tasks, indicating that existing gLMs have not learned strong protein/genomic sequence representations. We disagree that this means the datasets are ill-suited for gLMs.
> > >
> > > > It would be useful to include the foundation models that are not transformer based, such as Caduceus (a state-space model)
> > >
> > > This is a great point and we would like to clarify that our baselines already include the non transformer-based model Evo (hyena SSM based). Regarding the reviewer's suggestion of including Caduceus, we note that the Caduceus model is trained only on the human reference genome. Because our benchmark does not include human genomic sequences (as we prioritize diverse evaluation across all organisms, especially non-model organisms like archaea) in this case it does not make sense to include Caduceus as a baseline model. To our knowledge, there are no other widely used non-transformer pLMs or gLMs besides Evo.
> > >
> > > > Since it appears to be the case that no single model performs well across all tasks, the authors should more explicitly guide ML practitioners on the use cases of the tasks.
> > >
> > > We agree that no single model performs best across all tasks (as found in the analogous MTEB benchmark for LLMs), and because of this, practitioners should choose the model that performs best for their specific downstream use-case. Regarding the differentiation between classification and clustering tasks, the tasks are formulated based on the label type, and the quantity of available labels (datasets with scarce labels need to be formulated as zero-shot clustering/retrieval tasks).

---

### Official Review · Reviewer_tQZv · 2024-10-31

**Soundness:** 2
**Presentation:** 2
**Contribution:** 2
**Rating:** 5
**Confidence:** 4

**Summary:**

The authors propose a novel benchmark, diverges genomic embedding benchmark (DGEB) which proposes embedding evaluation across a variety of tasks across a diverse set of species. The authors posit that previous evaluations consisting primarily of DMS data and biophysical property evaluation is insufficient. Instead they propose an evaluation across different modalities consisting of RNA, and Protein allowing investigation of embedding qualities across different modalities. The authors have an extensive focus on investigating interacting subunits of specific protein families across different phylogenetic branches. Using this dataset the authors investigate capabilities of genomic and and protein language models. They assess impacts of scaling and comparing across modality performance.

**Strengths:**

- The authors perform homology deduplication when splitting the dataset based on sequence similarity. This is an important step to avoid information leakage.
- The authors are able to perform evaluation across different modalities benchmarking capabilities of current genomic language models against pLMs.
- The authors expand evaluation beyond eukaryotes and extensive assess representation qualities of sequences from archea and bacteria.

**Weaknesses:**

- The authors don't propose simple baselines such as supervised models or naive approaches for any of the tasks.
- One of the gaps identified is function takes place across diverse scales. There is no SNP evaluation and most of the evaluation consists of assessing homology relationships between sequences across different species or within.
- Overall I think a double blind venue isn't the right setting for evaluation of dataset papers since it requires assessment of the codebase. In addition it requires deep domain expertise in the subject area which is impossible to assess in a double blind setting. Out of the 4 design principles of DGEB it is impossible to assess 3 consisting of simplicity, extensibility and reproducibility. I think the quality of this work would be best assessed in a journal or a datasets and benchmarks track.
- The authors aren't very precise with their vocabulary:
	- "For BiGene Mining, we curated functionally analogous sequences found in two phylogenetically distant taxa (e.g. Bacteria and Archaea) or interacting paralog pairs in sets of orthologous sequences. "
	- Paralogs are homologous sequences related through duplication events. Orthologs are sequences related through speciation events. When the authors describe interacting genes do they mean physically interacting? If so what kind of evidence is used to support this claim?

**Questions:**

- "this metric cannot be used to determine how well a model can abstract evolutionary and functional relationships between non-homologous proteins."
	- I don't really understand what the authors mean by this sentence. Typically the goal of a benchmark is to assess function from sequence why would we want to abstract it?
- "These are important properties, they are too coarse in scope to evaluate whether a model has learned biologically meaningful functional information."
	- Is there evidence to support this claim?
- The authors argue that there is currently a eukaryotic bias in the evaluations available. I wonder if it's such a bad thing since most of the applications that we are interested in have to do with perturbing eukaryotic organisms?
- Another question that is important to mention in any sequence based evaluation benchmark  is that biotechnology is continuously improving and some of the measurement methodologies currently are flawed. What is the role of batch effects in constructing these datasets? How will these evaluation datasets continue to improve overtime as our ability to measure cell properties keeps improving?

---

> ### Author Response · Authors · 2024-11-22
> **Response to reviewer tQZv Part 1**
>
> Thank you for the thoughtful review, we appreciate you highlighting DGEB's ability to handle different modalities and inclusion of diverse sequences beyond eukaryotes.
>
> > The authors don't propose simple baselines
>
> Thank you for raising this point. We have added a one-hot baseline, where the sequence is represented as one-hot vector per residue, and mean-pooled across the sequence dimension like all benchmarked models. The baseline results are shown in Figure 3 as dotted lines.
>
> > There is no SNP evaluation and most of the evaluation consists of assessing homology relationships between sequences across different species or within.
>
> Thank you for pointing this out. We acknowledge that the current evaluation suite in DGEB focuses on higher level representations (single-element, inter-element and multi-element) as we identified this to be a major gap in existing benchmarks. DGEB complements existing residue-level benchmarks (e.g. ProteinGym, BEND) that focus on token/residue-level representations. We believe that it would be valuable to incorporate more residue-level benchmarks to our suite in the future. We have added this point to the Limitation section: "DGEB's current evaluation suite focuses on single-element, inter-element, and multi-element scales of representations, and is designed to complement existing benchmarks that focus on residue-level representations (e.g. mutational effects (Notin et al 2023, Marin et al 2024)."
>
> >  Overall I think a double blind venue isn't the right setting for evaluation of dataset papers since it requires assessment of the codebase. I think the quality of this work would be best assessed in a journal or a datasets and benchmarks track.
>
> We have added the anonymized code as a zip file in the OpenReview supplementary data. This paper is submitted under Primary Area: "Datasets and Benchmarks" track.
>
> > The authors aren't very precise with their vocabulary: "For BiGene Mining, we curated functionally analogous sequences found in two phylogenetically distant taxa (e.g. Bacteria and Archaea) or interacting paralog pairs in sets of orthologous sequences. " Paralogs are homologous sequences related through duplication events. Orthologs are sequences related through speciation events. When the authors describe interacting genes do they mean physically interacting? If so what kind of evidence is used to support this claim?
>
> Thank you for pointing this out. We recognize that this language is confusing. We have edited this sentence as "For BiGene Mining, we curated functionally analogous sequences found in two phylogenetically distant taxa (e.g. Bacteria and Archaea) or orthologs of interacting protein pairs across many species." The dataset we curated for this task is now renamed as "ModBC BiGene mining", where the physically interacting pairs of ModB and ModC are curated, and the task is to find the interacting ModC for a given ModB, in a set of ModC orthologs (ModCs found in different species).
>
> > "this metric cannot be used to determine how well a model can abstract evolutionary and functional relationships between non-homologous proteins." I don't really understand what the authors mean by this sentence.
>
> Thank you for flagging this. We have edited this sentence to "this metric cannot be used to determine how well a model can represent evolutionary and functional relationships between non-homologous proteins." The context of this sentence was to highlight that while DMS benchmarks are appropriate for assessing fitness effects of mutations in a homologous set of sequences, they do not assess how well a model represents relationships between distant and non-homologous sets of sequences.
>
> > "These are important properties, they are too coarse in scope to evaluate whether a model has learned biologically meaningful functional information." Is there evidence to support this claim?
>
> We have modified this sentence to "These are important properties, they are too coarse in scope to evaluate whether a model has learned more granular functional information (e.g. enzymatic function, protein interaction)".

---

> > ### Comment · Reviewer_tQZv · 2024-11-22
> >
> > Overall, I found the writing in this paper to be imprecise. While the authors have made an effort to address this in the rebuttal, there are still areas that require significant improvement. In my view, Appendix A is one of the most important parts of the paper, as it provides additional justification and explanations regarding the methodology used for dataset curation. However, the explanations often feel incomplete, and methodological choices are insufficiently detailed. For instance, processing choices such as:
> >
> > - Orthologous genes were identified through manual inspection.
> > - Labeled MopB family sequences were obtained from sequences used to construct Figure 1 in Wells et al. (2023).
> >
> > raise more questions than they answer.
> >
> > Additionally, the naive baseline appears to be treated as an afterthought, despite being a crucial component of any benchmarking paper—particularly in a field where the utility of foundation models remains a topic of debate. I was unable to find detailed information about the parameters or architecture of this baseline, aside from the fact that it was trained on one-hot encoded sequences.
> >
> > I believe this is an important work, but given its interdisciplinary nature, extra care must be taken to convincingly establish the validity of the benchmark for both the biological and machine learning communities. The authors could strengthen this paper by providing more comprehensive explanations for each benchmark, clearly articulating the methodology and reasoning behind their inclusion. Furthermore, a more detailed discussion of the baseline method should be incorporated. Commonly used methods such as CNNs and possibly even a classical ML method trained on featurized kmer vectors.
> >
> > In its current form, I am not fully convinced this paper is a clear accept. However, I am willing to defer to the judgment of the other reviewers if they feel strongly about its merit.

---

> > > ### Author Response · Authors · 2024-11-24
> > > **Response to Reviewer tQZv**
> > >
> > > Thank you for providing this additional feedback, and we completely agree that "extra care must be taken to convincingly establish the validity of the benchmark for both the biological and machine learning communities." As such, we have carefully curated each dataset to be of high biological relevance, with "special attention paid to include non-representative species and ensuring that train-test leakage is minimized" (Reviewer zUhM).
> > >
> > > We aim to be as detailed as possible in the description of the tasks (Section 3.2),  datasets (Section 3.3) and curation methodology (Appendix A). Regarding the two issues you mention, the methodological details were contained in the relevant sections, but we agree that the clarity could be improved:
> > >
> > > > Orthologous genes were identified through manual inspection.
> > >
> > > Thank you for bringing this to our attention. While the original text outlined each step of the manual inspection after this statement, we agree that this was not sufficiently clear, and we updated using the phrasing below:
> > >
> > >
> > > "RefSeq annotations were obtained for E. coli str. K-12 substr. MG1655 (GCF 000005845.2) and Sulfolobus acidocaldarius (GCF 000012285.1). Orthologous genes were identified as follows: 1) Genes with exactly matching annotations were identified first and added to the dataset; numerous genes with nearly identical, but not exactly matching, annotations, were also added to the dataset. 2) Genes without highly similar annotations and with matching function indicated through other databases such as UniRef, were marked as orthologs. 3) Unannotated genes in Sulfolobus were identified as orthologs to E. coli sequences through a combination of genome context information, matching HMM domains, and high structural similarity identified through a Foldseek (van Kempen et al., 2024) search between the predicted structure of Sulfolobus sequences against the structures available for E. coli MG1655; 4) For Sulfolobus genes with ambiguous RefSeq annotations, at least two such clues (matching UniRef annotations, genome context clues, matching HMM annotations, and Foldseek structural similarity) were required to assign an orthologous pair. 5) Genes where multiple homologs existed in both E. coli and Sulfolobus genomes were deliberately excluded."
> > >
> > > > Labeled MopB family sequences were obtained from sequences used to construct Figure 1 in Wells et al. (2023).
> > >
> > > We agree that this could have been phrased in an improved way and provide context for why we chose to use Wells et al (2023) as reference. We have updated to the following:
> > >
> > > "Labeled MopB family sequences, displayed in the phylogenetic tree of Figure 1 in Wells et al. (2023), were obtained from their provided Supplementary Materials (https://itol.embl.de/tree/249112161424681659917609). Wells et al (2023) conducted one of the  most comprehensive and up-to-date classification of MopB family enzymes to date."
> > >
> > >
> > > > Additionally, the naive baseline appears to be treated as an afterthought, despite being a crucial component of any benchmarking paper—particularly in a field where the utility of foundation models remains a topic of debate. I was unable to find detailed information about the parameters or architecture of this baseline, aside from the fact that it was trained on one-hot encoded sequences.
> > >
> > >
> > > We agree that the baseline is a crucial component of a benchmark paper, and we thank the reviewers for requesting the baseline, as this significantly strengthened the paper. However, we disagree that the baseline is naive or an afterthought. Our benchmark evaluates vector representations of biological sequences, and the one-hot baseline representation (after mean-pooling) is equivalent to amino-acid/nucleic-acid composition. This is the most natural featurization of biological sequences, and this representation has been shown to be highly predictive of challenging biological tasks such as transmembrane β-barrel protein identification [1] and microbial growth conditions [2]. We agree that additional information on the baseline should be provided, and we have added Appendix H with details. We clarify that the one-hot representation is treated the same way as all other model representations:
> > > - For supervised tasks (EC, Convergent Enzyme, MIBiG classification), the representations are mean-pooled and fed to a Logistic regression classifier or k-NN classifier(see Section 3.2)
> > > - For unsupervised tasks, the representations are mean-pooled, and evaluated using the method specified in Section 3.2 (such as clustering, retrieval, etc.)
> > >
> > >
> > > We hope these clarifications regarding the methodology and baseline address the reviewers concerns.
> > >
> > > [1] https://pmc.ncbi.nlm.nih.gov/articles/PMC1274253/
> > >
> > > [2] https://www.biorxiv.org/content/10.1101/2024.03.22.586313v1.full.pdf

---

> > > > ### Comment · Reviewer_tQZv · 2024-11-24
> > > >
> > > > The current baselines could be improved. Using a distribution over nucleotide or amino acid composition isn't sufficient, especially for evaluating foundation models. A stronger benchmarking would include a dilated CNN model for tasks with labels and k-mer counts as a baseline for tasks without labels.
> > > >
> > > > This concern goes beyond the two examples I provided in an earlier post regarding Appendix A. The section as a whole doesn’t feel thorough enough, and it could benefit from more care in explaining the individual tasks and the reasoning behind them. For example, in the case of ModBC, the context the authors provided in their rebuttal isn’t reflected in the original entry, which makes it harder to fully understand its inclusion.
> > > >
> > > > I’d recommend going through each dataset carefully, clearly outlining the motivation for including it and the methods used to generate it. This would make the paper much stronger and easier to follow.
> > > >
> > > > As it stands, though, the gaps make it hard for me to adjust my score at this time.

---

> > > > > ### Author Response · Authors · 2024-12-02
> > > > >
> > > > > Thank you for the followup and additional suggestions.
> > > > >
> > > > > > A stronger benchmarking would include a dilated CNN model
> > > > >
> > > > > Considering our benchmark is designed to evaluate pretrained model embeddings with zero-shot or linear projection, we believe that a supervised CNN baseline does not provide a fair comparison.
> > > > >
> > > > > >  k-mer counts as a baseline for tasks without labels
> > > > >
> > > > > We use amino-acid/nucleic-acid composition as the baseline (equivalent to 1-mers) which is standard for biological sequence representations, as explained in [1], where they "restrict features to 1-mers because using longer k-mers of DNA or amino acid sequences can introduce spurious phylogenetic correlations due to their intrinsic phylogenetic signal and their high-dimensionality relative to the amount of training data available"
> > > > >
> > > > > > Appendix A could benefit from more care in explaining the individual tasks and the reasoning behind them.
> > > > >
> > > > > Thank you for suggesting this, we recognize that providing additional biological context would be valuable for ML practitioners,   and we can these details in the camera ready (as the manuscript update deadline has passed).
> > > > >
> > > > > [1] https://www.biorxiv.org/content/10.1101/2024.03.22.586313v1.full.pdf

---

> ### Author Response · Authors · 2024-11-22
> **Response to reviewer tQZv Part 2**
>
> > The authors argue that there is currently a eukaryotic bias in the evaluations available. I wonder if it's such a bad thing since most of the applications that we are interested in have to do with perturbing eukaryotic organisms?
>
> We argue that "Labels are heavily biased towards model organisms (e.g. Human), therefore performance on species-specific evaluation tasks are not guaranteed to transfer to other organisms." Most of biotechnological breakthrough discoveries in the past decades were made in non-human systems (e.g. CRISPR-Cas (bacteria), RNAi (plants and worms), DNA Polymerase I for PCR (bacteria), GFP (Jellyfish), Antibiotics (bacteria)). Models that overfit on Human genomic evaluation sets will not contribute to enabling future high-impact discoveries in biotechnology that are likely to be found in diverse organisms.
>
> > Another question that is important to mention in any sequence based evaluation benchmark is that biotechnology is continuously improving and some of the measurement methodologies currently are flawed. What is the role of batch effects in constructing these datasets? How will these evaluation datasets continue to improve overtime as our ability to measure cell properties keeps improving?
>
> This is a great point. We believe that evaluation benchmarks must continually improve over time. We explicitly mention and implement versioning: "We version both the software and the datasets and include versioning in the results, making the benchmark results fully reproducible." We also explicitly built a platform that can be improved over time: "[...] existing functional annotations must be continuously refined and expanded. DGEB supports simple extension of tasks and datasets. New or revised datasets can be uploaded to the HuggingFace Hub and new evaluation tasks can easily be added through GitHub pull requests." We expect there to be batch effects across versions as the granularity and accuracy of our sequence-to-function mapping improve. These batch effects are inevitable and will be tracked with thorough versioning of both the benchmarking software and datasets.

---

### Official Review · Reviewer_zUhM · 2024-11-04

**Soundness:** 4
**Presentation:** 3
**Contribution:** 3
**Rating:** 6
**Confidence:** 4

**Summary:**

The authors present the Diverse Genomic Embedding Benchmark (DGEB), an embedding benchmark inspired by benchmarks for natural language embeddings. DGEB includes six different tasks (BiGene mining, classification, pair classification, evolutionary distance similarity, clustering, and retrieval) across 18 expert-curated datasets selected for phylogenetic diversity, difficulty, and variety of biological significance. Notably, DGEB works for any datasets that generate embeddings, including both nucleotide and amino acid variants, causal and masked models. Special attention is paid to including non-representative species and ensuring that train-test leakage is minimized. The package is open-sourced and designed to be easy to use, including a public leaderboard to determine the models with the best embeddings performance; it can also be extended by adding further datasets to the HuggingFace hub. The authors start off by benchmarking ESM-2, ESM-3, ProGen2, ProtTrans, Nucleotide Transformer, and Evo, and do not identify a single best model: they do find, however, that many tasks show performance saturation at low accuracies, suggesting there is something inadequate in the way that current models are trained.

**Strengths:**

* **Benchmark design.** The benchmarks in the paper are, in general, carefully thought out and sensible. Here are some particular highlights:
  * **Task diversity.** The paper covers a variety of biologically important benchmark tasks (rather than, for instance, just focusing on classification).
  * **Phylogenetic diversity.** The authors take great care to expand the benchmark tasks to include a diverse set of sequences from across the tree of life, allowing us to evaluate the extent to which genomic models generalize to underrepresented clades.
  * **Leakage considerations.** By deduplicating sequences in the benchmark datasets, and enforcing a 10% nucleotide identity cutoff between train and test sets, the authors minimize the risk of train-test leakage in datasets. This is a crucial, biologically-informed modification to conventional NLP embeddings benchmark that needs to be enforced in genomic contexts.
* **Modularity and usability.** The DGEB is designed to be easy to run on any model that generates embeddings, has a community leaderboard, and can even be extended with new benchmarks on the Huggingface Hub.
* **Comparison between AA and NA modalities.** This benchmark dataset allows us to compare and evaluate NA and AA modalities on four datasets. To the best of my knowledge, this is the first benchmark to facilitate such a comparison.

**Weaknesses:**

* **Evaluating base models**. While it seems reasonable to assume that more capable models will have geometries that more closely match various problems, this seems far from guaranteed. It could also be the case that more capable models fine-tune better, or even that more capable models have embeddings that more closely match the geometry of individual problems *in a specific subspace* (in which case one would need to learn the correct projection to make use of their embeddings). I realize addressing these possibilities is out of scope, but a more explicit discussion of these limitations is warranted.
* **Distance-based evaluations**. Many of the benchmark tasks make assumptions about the metric structure of the embedding space. There is some arbitrariness/inconsistency in what distance function is used where; in practice, the correct distance function is likely to depend on the specifics of the model architecture (e.g. Euclidean distances are not consistent in transformer models that use LayerNorm). Here are the different distance functions employed:
  * BiGene mining uses cosine similarity to a query gene
  * Evolutionary distance similarity uses a "kitchen sink" approach, comparing cosine, Euclidean, and Manhattan distances of embeddings to ground-truth distances (defined as the branches of a phylogenetic tree)
  * Multiclass, multi-label classification uses $k$-nearest neighbors classifiers, which assume a distance function (which distance function is used here? Presumably Euclidean?)
  * Pair classification uses cosine similarities, as well as a bunch of other distance metrics
    * Side note: it's a bit unclear exactly how the pair classification works. It seems that several distance functions are tried, but cosine similarity is privileged as the main metric---am I correct in my assessment?
  * Clustering uses $k$-means, which presumably uses Euclidean distance to assess the distance to each centroid
  * Retrieval uses cosine similarity
* **Fundamental limitation of using Euclidean embeddings**. For evolutionary distances, Euclidean space will be unable to accommodate the exponential growth of neighborhoods (this has motivated the development of hyperbolic embeddings for evolutionary distance estimation). The theoretical minimum distortion of phylogenetic distances is nonzero for Euclidean embeddings (and depends inversely on dimensionality)—thus, a benchmark like this is inherently limited. To this end, the authors could do a few things:
  * Acknowledge the fundamental limitations of evaluating evolutionary distances with Euclidean embeddings, and argue for the usefulness of their benchmark as a proxy/benchmark in spite of this.
  * Characterize the effect of the embedding dimension on evolutionary distance estimation accuracy.
* **Layer choice.** Another limitation of their method is the arbitrary choice of final/middle-layer embeddings. It is not clear which layer is chosen as "mid" when the size of the model fluctuates; I also wonder how influential this choice of "mid" layer is on model performance (e.g. does choosing layer 8 versus 9 make a substantial difference? How does this interact with model choice)
* **Results presentation.** The presentation of the results across many subfigures, rather than a single benchmark table, is somewhat confusing. I recommend cleaning up the table in Appendix G and including it as the main result. (This may be more of a matter of personal preference)
* **Justification.** The authors have done a great job of motivating their choice of datasets and benchmark tasks. In light of the above comments, I would like to see the authors justify the specific design choices that go into the *architectures* they use as well, and comment on the limitations of these approaches.

**Questions:**

* Are there any regression tasks that can be included in this benchmark? What would a reasonable, general framework for training and evaluating a simple regression model on top of these embeddings look like?
* Same question, but for link prediction.
* Is it possible to measure/quantify, or at least demonstrate, the effect of not enforcing the phylogenetic train-test split?

---

> ### Author Response · Authors · 2024-11-22
> **Response to reviewer zUhM Part 1**
>
> Thank you for your review. We are glad you find the design of DGEB to be "carefully thought out and sensible" and the direct comparison between AA and NA modalities to be a useful contribution to the field.
>
> > Assumption that more capable models will have geometries that more closely match various problems
>
> This is a great point, we agree that there is no guarantee that models will have geometries matching specific problems, and that this is an assumption based on empirical evidence. We have added this to the Limitations section:
> "DGEB includes multiple zero-shot tasks, as ground-truth labels for biological function are sparse and biased. These tasks rely on embedding geometry to evaluate model performance. The assumption that models capturing important features of biological function have geometry directly matching the given tasks is not guaranteed. Future research could explore methods for identifying and leveraging relevant subspaces within model embeddings."
>
> > Many of the benchmark tasks make assumptions about the metric structure of the embedding space, the correct distance function is likely to depend on the specifics of the model architecture
>
> We agree that there is not necessarily a single correct distance function for any task. Because of this, we default to the commonly accepted cosine similarity as the metric for all tasks relying on embedding distances, except for evolutionary distance tasks, where there is less certainty in the best metric so we select the one with highest Pearson correlation. For clustering, we use k-means with euclidean distance, as other distance metrics are not supported by sklearn k-means implementation. Regarding pair classification, other distance metrics are computed, but only cosine similarity is selected as the main metric and the others are unused (will be removed in the next release).
>
> > Limitation of using Euclidean embeddings for phylogenetic distances
>
> Thank you for highlighting this, and we agree that Euclidean space is not optimal for representing phylogenetic tree structures. While Euclidean embeddings have limitations, they still provide a useful starting point for comparing model performance, and we believe this benchmark will be important for evaluating novel hyperbolic architectures, baselined by Euclidean embedding model results. We add this acknowledgment to the limitations section.
>
> > It is not clear which layer is chosen as "mid" when the size of the model fluctuates. I also wonder how influential this choice of "mid" layer is on model performance
>
> We evaluate all models using embeddings from the middle and last layer to account for layer specialization noted in previous studies (Section 4.2.1). The middle layer is extracted automatically for each model using its config (which specifies model depth). We do not benchmark each layer to avoid issues with multiple hypothesis testing, and to keep the evaluation fair over models with varying depth. Empirically, we do not observe discrete jumps in performance across  consecutive layers, likely due to the residual stream in modern transformer architectures.
>
> > The presentation of the results across many subfigures, rather than a single benchmark table, is somewhat confusing. I recommend cleaning up the table in Appendix G and including it as the main result. (This may be more of a matter of personal preference).
>
> We chose to represent the results in the main text as Figure 3 instead of the table in Appendix G, because the figure provides immediate ability to compare scaling performance across parameter count. The table cannot show the continuous axis of model scale, and is provided as reference to readers.
>
> >  I would like to see the authors justify the specific design choices that go into the architectures they use as well
>
> We would like to clarify that our benchmark focuses on evaluating publicly available pre-trained models. We did not train any new models in this study. Our primary objective is to assess existing models on biological function. Justifying the design of these pre-trained architectures is beyond the scope of our current work.

---

> ### Author Response · Authors · 2024-11-22
> **Response to reviewer zUhM Part 2**
>
> > What would a reasonable, general framework for training and evaluating a simple regression model on top of these embeddings look like?
>
> The DGEB benchmark is designed to be easily extendable with community contributions of datasets and tasks. The addition of regression tasks, like gene expression, would be very similar in implementation to the existing classification tasks, like EC Classification, which trains a sklearn LogisticRegression on the embeddings for the provided train/test split. In this case a LinearRegression layer should be used.
>
> > Same question, but for link prediction.
>
> DGEB includes tasks which are considered link prediction, such as the operon prediction tasks which identify interacting pairs from non-interacting pairs. This could be extended to additional link prediction tasks with higher degree graphs, for example using nearest-neighbor link prediction.
>
> > Is it possible to measure/quantify, or at least demonstrate, the effect of not enforcing the phylogenetic train-test split?
>
> We would like to clarify that the phylogenetic evolutionary distance  similarity (EDS) task does not use a train-test split, as it is a zero-shot task.

---

> > ### Comment · Reviewer_zUhM · 2024-11-26
> >
> > Thank you for your detailed response to my questions. I appreciate your adding nuance to the revised draft in accordance with our discussion of the meaning of inner products and distances for genome embeddings, as well as your patient elaboration of various points of confusion in my questions. I am still not convinced that this embeddings benchmark approach carries over accurately to genomic datasets, and have concerns around the impact that the adoption of DGEB as a community standard will have on practitioner's thinking about the geometry of genomic representations. Because of this, I am keeping my score at a 6.

---

### Meta-Review · Area_Chair_kGD5 · 2024-12-21

**Metareview:**

In this work, a novel benchmark is presented for evaluating language models trained on proteins and genomic sequences, across a variety of tasks over sequences curated across many diverse species.

While novel rigorous benchmarks in this space are highly welcome, it is apparent that the Reviewers are split on whether to accept the work, and even the Reviewers with a positive final assessment of the work have expressed tangible concerns about some aspects of the data and how its adoption might be received by practitioners.

These concerns need to be addressed (in all cases, by bare minimum, through careful discussion -- and ideally through additional exprimentation / baselines where relevant) before I would be comfortable recommending the paper for acceptance. It is my opinion that these changes are not minor enough to justify accepting the paper at this point. The Reviewers did not oppose this decision.

**Additional Comments On Reviewer Discussion:**

One of the Reviewers called out that a previous revision of the supplementary code has accidentally de-anonymised the Authors' identities.

While this is on its own sufficient reason to reject the work, I have decided to ignore it as a factor in making my decision, mainly because it looked like an honest oversight by the Authors and the issue was contained to the code repository.

---

### Decision · Program_Chairs · 2025-01-22

Reject